# Implicit Neural Representations for Robust Joint Sparse-View CT Reconstruction

**Jiayang Shi***  
*Leiden Institute of Advanced Computer Science*  
*Leiden University*
*j.shi@liacs.leidenuniv.nl*

**Junyi Zhu***  
*Center for Processing Speech and Images*  
*KU Leuven*
*junyi.zhu@esat.kuleuven.be*

**Daniël M. Pelt**  
*Leiden Institute of Advanced Computer Science*  
*Leiden University*
*d.m.pelt@liacs.leidenuniv.nl*

**K. Joost Batenburg**  
*Leiden Institute of Advanced Computer Science*  
*Leiden University*
*k.j.batenburg@liacs.leidenuniv.nl*

**Matthew B. Blaschko**  
*Center for Processing Speech and Images*  
*KU Leuven*
*matthew.blaschko@esat.kuleuven.be*

**Reviewed on OpenReview:** *https://openreview.net/forum?id=XCzuQIOoXR*

## Abstract

Computed Tomography (CT) is pivotal in industrial quality control and medical diagnostics. Sparse-view CT, offering reduced ionizing radiation, faces challenges due to its under-sampled nature, leading to ill-posed reconstruction problems. Recent advancements in Implicit Neural Representations (INRs) have shown promise in addressing sparse-view CT reconstruction. Recognizing that CT often involves scanning similar subjects, we propose a novel approach to improve reconstruction quality through joint reconstruction of multiple objects using INRs. This approach can potentially utilize the advantages of INRs and the common patterns observed across different objects. While current INR joint reconstruction techniques primarily focus on speeding up the learning process, they are not specifically tailored to enhance the final reconstruction quality. To address this gap, we introduce a novel INR-based Bayesian framework integrating latent variables to capture the common patterns across multiple objects under joint reconstruction. The common patterns then assist in the reconstruction of each object via latent variables, thereby improving the individual reconstruction. Extensive experiments demonstrate that our method achieves higher reconstruction quality with sparse views and remains robust to noise in the measurements as indicated by common numerical metrics. The obtained latent variables can also serve as network initialization for the new object and speed up the learning process.[1]

---

*Equal contribution.

[1]We have used ChatGPT provided by OpenAI to assist in writing. The language model was employed at the sentence level for tasks such as fixing grammar and rewording sentences. We assure that all ideas, claims, and results presented in this work are human-sourced.

# 1 Introduction

Computed Tomography (CT) is a crucial non-invasive imaging technique, extensively employed in medical diagnostics and industrial quality control. In CT, an object's internal structure is reconstructed from X-ray projections captured at multiple angles, posing a complicated inverse problem. In certain situations, reducing the number of CT measurements can yield advantages such as decreased radiation exposure and enhanced production throughput. However, this sparsity in measurements, along with other factors such as the presence of noise and the size of detectors, complicates the reconstruction process, making it an ill-posed inverse problem. Such challenges arise not only in CT reconstruction but also across diverse computational tasks. Therefore, while our study centers on sparse-view CT reconstruction, the core ideas are transferable to numerous inverse problems, such as Magnetic Resonance Imaging and Ultrasound Imaging.

Different strategies have been developed to address the challenges of sparse-view CT reconstruction by incorporating auxiliary information. Supervised learning techniques learn mappings from measurements to images (Zhang et al., 2018b; Han & Ye, 2018; Zhu et al., 2018; Wu et al., 2021). These methods train on extensive datasets containing dense measurements, synthesizing the sparse view through physical models of the measurement process. Alternatively, Song et al. (2022) introduces an approach that directly learns the image distribution from high-quality reconstructed images using generative models. Several diffusion-based approaches (Chung et al., 2022; Liu et al., 2023; Song et al., 2023; Xu et al., 2024) have been specifically tailored for CT applications, further refining the foundational ideas presented by Song et al. (2022). However, their method still relies on a large, domain-specific dataset, that closely matches the target of reconstruction. Dependence on large datasets introduces practical limitations to supervised learning techniques and generative methods, particularly when collecting large-scale and high-quality data is intractable, such as due to privacy regulation in medical cases or timeliness in industrial settings. It is important to note that supervised learning and diffusion-based approaches represent only a subset of the methods available for CT reconstruction. There are also some studies that leverage heuristic image priors, e.g. Total Variation (TV) (Sidky & Pan, 2008; Liu et al., 2013; Zang et al., 2018). While effective, these methods lack the ability to incorporate domain-specific enhancement. Another approach involves using a few dense-view images as priors for reconstruction (Chen et al., 2008; Shen et al., 2022). However, this method hinges on the availability of dense-view images similar to the target object, which is often impractical when the appearance of the object to be reconstructed is unknown.

Building on the foundations of CT reconstruction, many works explore the potential of implicit neural representations (INRs). Thanks to the advantages of INRs, such as their continuous representation nature in contrast to conventional discrete representations (Lee et al., 2021; Xu et al., 2022; Grattarola & Vandergheynst, 2022), these methods have consistently delivered promising reconstruction results with limited number of measurements (Zang et al., 2021; Zha et al., 2022; Rückert et al., 2022; Wu et al., 2023b). Given INRs' proven capabilities in CT reconstruction and the known advantages of incorporating prior information, either heuristically through TV (Zang et al., 2021) or explicitly from dense-view CT images (Shen et al., 2022), we seek to merge these two paradigms. Aiming for broad applicability, we avoid reliance on pre-curated datasets. Instead, we draw inspiration from the fact that modern CT machines routinely scan similar subjects, such as patients in hospitals or analogous industrial products. This commonality drives us to explore a novel question in this research:

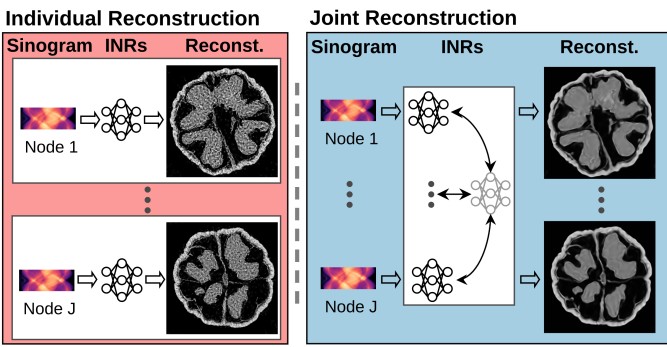

Figure 1: Compared to individual reconstruction, joint reconstruction enables INRs to share statistical regularities among multiple objects, thereby enhancing the quality of reconstruction under conditions of sparse-view CT scans and inherent measurement noise. The examples shown here are CT reconstructions of walnuts from a sparse set of projection angles, with the joint reconstruction results obtained using our proposed approach.

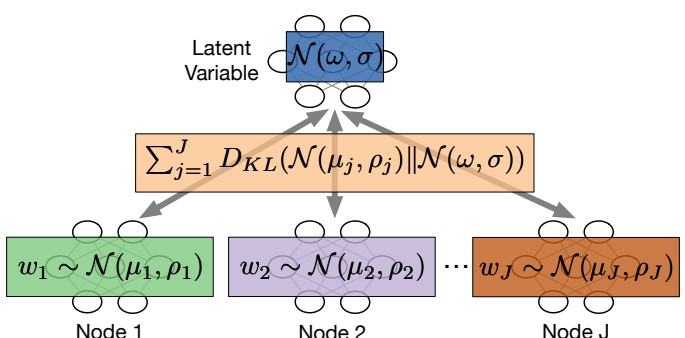

Figure 2: Framework of our proposed method. It uses latent variables to capture the relation among all reconstruction nodes. The latent variables are updated based on all nodes and regularize each individual reconstruction via minimizing the KL divergence terms $D_{KL}$. $\boldsymbol{w}_j$ denotes the parameters of j-th node, distributed according to $\mathcal{N}(\boldsymbol{\mu}_j, \boldsymbol{\rho}_j)$, while $\mathcal{N}(\boldsymbol{\omega}, \boldsymbol{\sigma})$ specifies the prior distribution of parameters $\boldsymbol{w}_{1:J}$.

*Can INRs use the statistical regularities shared among multiple objects with similar representation to improve reconstruction quality through joint reconstruction?*

Figure 1 contrasts the methodologies of individual reconstruction against joint reconstruction. In our exploration of this research avenue, we found that several existing methods can be adapted for our purpose (Zhang et al., 2013; Ye et al., 2019; Tancik et al., 2021; Martin-Brualla et al., 2021; Kundu et al., 2022). These approaches have utilized the inherent statistical regularities across different objects, represented by the network weights in INRs. However, they often focus on addressing issues like convergence rate (Tancik et al., 2021; Kundu et al., 2022) rather than optimizing the reconstruction quality itself. As demonstrated in our experimental evaluations §5, these methods achieve lower PSNR and SSIM compared to our proposed method in joint reconstruction scenarios.

To address our research question, we propose a novel INR-based Bayesian framework, specifically tailored to adaptively integrate prior information into network weights during the training process. This is achieved by utilizing latent variables that encapsulate commonalities in the parameter space among multiple objects' neural representations. The latent variables representing prior information are then leveraged to enhance the accuracy of individual reconstructions. The convergence of our model is guided by minimizing the Kullback-Leibler (KL) divergence between the prior and the estimated posterior distributions of the neural representation networks. Importantly, our framework can automatically adjust the regularization strength of the prior information based on the similarity among neural representation networks. Overall, our framework provides a robust solution to the challenges posed by sparse and noisy measurements, and varied reconstructions in CT imaging. An illustration of our proposed Bayesian framework is presented in Figure 2.

**Our Contributions:** i) We explore a novel problem of INR-based joint reconstruction in the context of CT imaging, supported by a comprehensive review of existing methods that could be adapted to address this challenge. ii) We propose a principled Bayesian framework that adaptively integrates prior information throughout the training process for enhanced reconstruction quality. iii) Through extensive experiments, we evaluate various facets of reconstruction performance using common numerical metrics. Our results establish that our method outperforms the compared INR-based baselines.

## 2  Related Work

We briefly outline key studies related to our focal areas, with a comprehensive understanding of INR and Neural Radiance Fields (NeRF) available in the survey (Tewari et al., 2022; Molaei et al., 2023). The key working principles of INR for CT reconstruction is explained in §3.

Coordinate-based Multi-Layer Perceptrons (MLPs) intake spatial coordinates and output values, such as RGB or density, processed through MLP. Unlike traditional discrete representations that directly map inputs to outputs on pixel grids, these coordinate-based outputs are generated by the MLP. Therefore, such coordinate-based representations are often regarded as INRs. Originally, INRs faced challenges in capturing high-frequency details. However, the introduction of coordinate encoding, such as Fourier feature transformations (Jacot et al., 2018; Tancik et al., 2020), has significantly enhanced their ability to represent finer details.

NeRF is a state-of-the-art INR approach that reconstructs scenes from photos taken at multiple angles (Mildenhall et al., 2021; Barron et al., 2021; 2022), similar to CT where projections are acquired at different

angles to compute reconstruction. While NeRF processes optical images to render RGB values by considering light transmittance and occlusion, CT reconstruction calculates attenuation coefficients to reveal internal compositions using penetrating X-rays. We therefore briefly introduce the NeRF methods that we adapt as comparison baselines. Building on the groundwork, several techniques have emerged to leverage the shared information across different objects or scenarios. NeRF-wild (Martin-Brualla et al., 2021) differentiated between static and transient scene aspects, an approach echoed in video representations (Li et al., 2021; Mai & Liu, 2022). Tancik et al. (2021) introduced meta-learning techniques to learn a meta initialization which converges faster than training from scratch. They have also applied their method to CT reconstruction. For the same purpose of accelerating convergence, Kundu et al. (2022) introduced federated learning to obtain the meta initialization. Chen & Wang (2022) proposed using transformers as the meta-learners.

INRs' potential in CT reconstruction has been exploited in various ways. Sun et al. (2021) focused on representing sparse measurements. Zang et al. (2021) combined INRs with TV and non-local priors for CT reconstruction. Notable advancements include cone-beam CT optimization (Zha et al., 2022) and adaptive hierarchical octree representation (Rückert et al., 2022). Wu et al. (2023b) proposed to improve reconstruction precision by reprojecting inferred density fields. Shen et al. (2022) proposed to pre-train the INR network with a high-quality prior image. Reed et al. (2021) incorporated a scene template and motion field in reconstructing time-varying CT. Furthermore, Wu et al. (2023a) explored unsupervised INRs for metal artifact reduction in CT imaging.

## 3  Problem Statement and Preliminaries

The CT acquisition process, in an ideal noiseless scenario, can be mathematically described by the equation: $\boldsymbol{y} = \boldsymbol{A}\boldsymbol{x}$, where $\boldsymbol{x} \in \mathbb{R}^m$ represents the unknown object of interest and $\boldsymbol{y} \in \mathbb{R}^n$ symbolizes the ideal noiseless measurements. In practice, the actual obtained measurements $\widetilde{\boldsymbol{y}} \in \mathbb{R}^n$ are often noisy, where the difference between actual and ideal measurements $\boldsymbol{\epsilon} = \widetilde{\boldsymbol{y}} - \boldsymbol{y}$ is the measurement noise. We note that the measurement noise $\boldsymbol{\epsilon} \in \mathbb{R}^n$ is a stochastic term representing general noise, including non-additive types like Poisson noise, which is the primary focus of this study. For simplicity and to maintain focus on the methodological aspects, this manuscript primarily expands formulas under the assumption of noiseless measurements $\boldsymbol{y}$, unless specified otherwise.

These measurements arise from the interaction between the measurement matrix $\boldsymbol{A} \in \mathbb{R}^{n \times m}$ and the object. The task in CT is to infer the unknown object $\boldsymbol{x}$ from the acquired CT measurements $\boldsymbol{y}$. The inherent challenge lies in the common sparsity of these measurements, resulting in $m > n$. This makes the reconstruction problem ill-posed. Meanwhile, measurement noise can further corrupt the reconstruction results.

The INR designed for CT reconstruction is a function $f_{\boldsymbol{w}} : \mathbb{R}^3 \to \mathbb{R}^1$ parameterized by $\boldsymbol{w}$. It maps the three-dimensional spatial coordinates of the object to its intensity. INR consists of two components, formulated as $f_{\boldsymbol{w}} = \mathcal{M} \circ \gamma$. Here, $\gamma : \mathbb{R}^3 \to \mathbb{R}^d$ servers as the position encoding (Tancik et al., 2020; Barron et al., 2022; Martel et al., 2021; Müller et al., 2022), while $\mathcal{M} : \mathbb{R}^d \to \mathbb{R}^1$ acts as the neural representation. Typically, $\mathcal{M}$ is an MLP.

The function $f_{\boldsymbol{w}}(\cdot)$ takes a coordinate $\boldsymbol{c}_i \in \mathbb{R}^3$ and maps it to the intensity value $v \in \mathbb{R}^1$. For a full set of coordinate $\boldsymbol{C} := \{\boldsymbol{c}_1, \boldsymbol{c}_2, \ldots, \boldsymbol{c}_N\}$, the INR outputs the representation of the entire object as $\mathcal{F}_{\boldsymbol{w}}(\boldsymbol{C}) := \{f_{\boldsymbol{w}}(\boldsymbol{c}_1), f_{\boldsymbol{w}}(\boldsymbol{c}_2), \ldots, f_{\boldsymbol{w}}(\boldsymbol{c}_N)\}$. The network's parameters $\boldsymbol{w}$ can be optimized by minimizing the loss function: $\ell(\boldsymbol{w}) := \|\boldsymbol{A}\mathcal{F}_{\boldsymbol{w}}(\boldsymbol{C}) - \boldsymbol{y}\|_2^2$.

**Joint Reconstruction Problem.** We aim to simultaneously recover $J$ objects $\boldsymbol{x}_{1:J}$ using their corresponding measurements $\boldsymbol{y}_{1:J}$ and measurement matrices $\boldsymbol{A}_{1:J}$. The joint reconstruction problem can be formulated as:

$$\boldsymbol{w}_{1:J}^* = \operatorname*{arg\,min}_{\boldsymbol{w}_{1:J}} \sum_{j=1}^{J} \ell_j(\boldsymbol{w}_j), \tag{1}$$

$$\ell_j(\boldsymbol{w}_j) := \|\boldsymbol{A}_j \mathcal{F}_{\boldsymbol{w}_j}(\boldsymbol{C}) - \boldsymbol{y}_j\|_2^2. \tag{2}$$

While this formulation allows for different measurement matrices for each object, we primarily consider scenarios where all objects share the same measurement matrix $\boldsymbol{A}$ in our experiments. A straightforward way

to solve Equation (1) is by optimizing the $J$ nodes separately, akin to individual reconstruction. However, we propose introducing a dynamic prior, which links all the models $\boldsymbol{w}_{1:J}$ during training and updates in response to their optimization. This Bayesian framework is designed to exploit shared statistical regularities among different objects, thereby enhancing the quality of the joint reconstruction.

## 3.1 Existing Methods Available for Joint Reconstruction

In our exploration of this research avenue, we found several existing methods, though originally designed for different problems, have utilized the statistical regularities of multiple objects. These methods may be adaptable to our research problem. In this section, we delve into these methods in greater detail. Empirical evaluations, as demonstrated in §5, suggest that some of these techniques can outperform the individual reconstruction approach. However, they do not fully exploit the statistical regularities to improve the reconstruction quality, which we discuss in the §4.

**Composite of Static and Transient Representations.** Martin-Brualla et al. (2021) introduce a composite representation approach NeRFWild, designed to manage variable illuminations and transient occluders in collective observations. While CT does not involve variable illumination, their concept of combining "static" and "transient" components can be adapted for our context, which we term `INRWild`.

Let $\mathcal{G}_{\boldsymbol{\phi}}$ represent the neural representation for the static component ($\boldsymbol{\phi}$ is the weights of MLP $\mathcal{G}$) and $\mathcal{H}_{\boldsymbol{w}}$ signify the transient component ($\boldsymbol{w}$ is the weights of MLP $\mathcal{H}$). For a given set of $J$ objects, each object-associated reconstruction node has its distinct transient network $\boldsymbol{w}_j$ and corresponding transient feature $\boldsymbol{b}_j$. In contrast, the static network $\boldsymbol{\phi}$ is shared across all nodes. The objective for this framework is formulated as:

$$\boldsymbol{\phi}^*, \boldsymbol{w}_{1:J}^*, \boldsymbol{b}_{1:J}^* = \underset{\boldsymbol{\phi}, \boldsymbol{w}_{1:J}, \boldsymbol{b}_{1:J}}{\arg\min} \sum_{j=1}^{J} \|\boldsymbol{A}_j \left(\mathcal{H}_{\boldsymbol{w}_j}\left(\boldsymbol{b}_j, \mathcal{G}_{\boldsymbol{\phi}}^{\setminus r}(\boldsymbol{C})\right) + \mathcal{G}_{\boldsymbol{\phi}}^r(\boldsymbol{C})\right) - \boldsymbol{y}_j\|_2^2. \tag{3}$$

The output of the static network $\mathcal{G}_{\boldsymbol{\phi}}$ is split into two components: $\mathcal{G}_{\boldsymbol{\phi}}^r(\boldsymbol{C})$, which represents the static intensity, and $\mathcal{G}_{\boldsymbol{\phi}}^{\setminus r}(\boldsymbol{C})$, which serves as intermediate features for the transient networks. The symbols $r$ and $\setminus r$ denote the division of outputs into these two components. Detailed explanation and framework schematic are in Appendix B.2. At its core, `INRWild` emphasizes training the static network $\boldsymbol{\phi}$, which embodies most learnable parameters, using aggregated losses. Concurrently, the individual representations, characterized by $\boldsymbol{w}_{1:J}$ and $\boldsymbol{b}_{1:J}$, are refined based on $\boldsymbol{\phi}$ that corresponds to the static part. $\boldsymbol{w}_{1:J}$, $\boldsymbol{b}_{1:J}$ and $\boldsymbol{\phi}$ are jointly optmized to achieve the composition of static and transient parts.

**Model-agnostic Meta-learning (MAML)**: Meta-learning aims to train a network in a way that it can quickly adapt to new tasks (Finn et al., 2017; Nichol et al., 2018; Fallah et al., 2020). Several INR-based works have employed `MAML` to obtain a meta-learned initialization, thereby accelerating the convergence or enabling model compression (Tancik et al., 2021; Lee et al., 2021). In the `MAML` framework, computational cycles are organized into "inner loops" and "outer loops", indexed by $k = 1 \dots K$ and $t = 1, \dots, T$ respectively. For each node $j = 1, \dots, J$, the networks $\boldsymbol{w}_{1:J}$ are initialized according to the meta neural representation $\boldsymbol{w}_{1:J}^{(0)} = \boldsymbol{\theta}$. These networks then undergo $K$ steps of inner-loop learning: $\boldsymbol{w}_j^{(k)} = \boldsymbol{w}_j^{(k-1)} - \eta \nabla_{\boldsymbol{w}_j} \ell_j(\boldsymbol{w}_j^{(k-1)})$, where $\eta$ is the inner-loop learning rate. After these $K$ steps, the meta network $\boldsymbol{\theta}$ updates as follows:

$$\boldsymbol{\theta}^{(t)} = \boldsymbol{\theta}^{(t-1)} - \alpha \frac{1}{J} \sum_{j=1}^{J} (\nabla_{\boldsymbol{w}_j})^T \ell_j(\boldsymbol{w}_j^{(K)}) \frac{\partial \boldsymbol{w}_j^{(K)}}{\partial \boldsymbol{\theta}}, \tag{4}$$

where $\alpha$ is the outer-loop learning rate. Equation (4) is a form that can be further specialized. After $T$ steps of outer-loop optimization, the meta-learned neural representation $\boldsymbol{\theta}^{(T)}$ serves as an effective initialization for individual reconstructions.

**Federated Averaging (FedAvg)**: Kundu et al. (2022) suggested to employ `FedAvg` (McMahan et al., 2017) as the optimization framework of the meta-learned initialization. Like `MAML`, `FedAvg` also consists of inner and outer loops. The inner loop is identical to `MAML`. Whereas, the outer loop simplifies the optimization by

averaging all individual networks, represented as $\boldsymbol{\theta} = \frac{1}{J}\sum_j \boldsymbol{w}_j^{(K)}$. Essentially, the meta network acts as the centroid of all networks. [2]

## 4 A Novel Bayesian Framework for Joint Reconstruction

In this section, we introduce `INR-Bayes`, our Bayesian framework for INR joint reconstruction.

**Motivation.** `NeRFWild` is proposed to address scene reconstruction challenges, such as reconstructing popular sightseeing sites from in-the-wild photos. These scenarios often involve a primary target object, like a landmark, amidst various transient elements such as pedestrians and changing light conditions. A method that uses a composition of static and transient components operates under the assumption that the core representations of the scene—those of the main object—substantially overlap, despite different viewpoints or transient changes. This assumption typically holds true in 3D scene reconstruction, where observations are collected from multiple perspectives of the same object. Our empirical findings on `INRWild` indicate such methods do not work efficiently in CT reconstruction and other image-level reconstruction tasks (discussed in §5). Meta-learned initialization methods, such as `FedAvg` and `MAML`, train a meta model to capture a high-level representation, which can then be promptly adapted to individual objects. Although the meta model can effectively extract statistical regularities among multiple objects, we observe that this prior information tends to be lost during the adaptation phase. This loss occurs as the adaptation relies solely on local measurements, leading to an overfitting issue, as we will discuss in §5. Overfitting is a notorious problem in iterative methods of CT reconstruction (Herman & Odhner, 1991; Effland et al., 2020), which renders these methods sensitive to hyperparameters, such as early stopping. This sensitivity consequently increases the difficulty of deployment.

To enhance the joint reconstruction process, we propose a principled Bayesian framework. Our method employs latent variables, denoted by $\{\boldsymbol{\omega}, \boldsymbol{\sigma}\}$, to capture commonalities among individual INR networks. These latent variables actively serve as references throughout the entire reconstruction process, guiding the training of individual networks.

**Definition and Notation.** We introduce distributions to the networks $\boldsymbol{w}_{1:J}$ for $J$ objects, and define latent variables $\{\boldsymbol{\omega}, \boldsymbol{\sigma}\}$ that parameterize an *axis-aligned multivariate* Gaussian prior $\mathcal{N}(\boldsymbol{\omega}, \boldsymbol{\sigma})$ from which the weights $\boldsymbol{w}_{1:J}$ are generated. Each node is assumed to be of the same size. A Gaussian distribution is a natural and effective choice for the weight distribution of the MLP such as INR networks (de G. Matthews et al., 2018). The latent variables collectively serve to capture the shared trends within the networks, effectively quantifying the mutual information across different objects. To simplify the model, we assume the *conditional independence* among all objects: $p(\boldsymbol{w}_{1:J}|\boldsymbol{\omega}, \boldsymbol{\sigma}) = \prod_{j=1}^{J} p(\boldsymbol{w}_j|\boldsymbol{\omega}, \boldsymbol{\sigma})$. This assumption of conditional independence allows us to decompose the variational inference into separable optimization problems, thereby facilitating more efficient parallel computing.

Given that the measurements of the objects $\boldsymbol{y}_1, \ldots, \boldsymbol{y}_J$ are mutually independent and that each network focuses on a specific object, the posterior distribution of network weights and latent variables can be derived using the Bayes' rule as $p(\boldsymbol{w}_{1:J}, \boldsymbol{\omega}, \boldsymbol{\sigma}|\boldsymbol{y}_{1:J}) \propto p(\boldsymbol{\omega}, \boldsymbol{\sigma}) \prod_{j=1}^{J} p(\boldsymbol{y}_j|\boldsymbol{w}_j)p(\boldsymbol{w}_j|\boldsymbol{\omega}, \boldsymbol{\sigma})$. While this posterior enables various forms of deductive reasoning such as reconstruction uncertainty quantification, inferring the true posterior is often computationally challenging. Moreover, the selection of an appropriate prior $p(\boldsymbol{\omega}, \boldsymbol{\sigma})$ poses its own difficulties (Wenzel et al., 2020; Fortuin et al., 2022). To tackle these issues, we present an algorithm that aims at maximizing the marginal likelihood $p(\boldsymbol{y}_{1:J}|\boldsymbol{\omega}, \boldsymbol{\sigma})$, integrating out $\boldsymbol{\omega}_{1:J}$. Detailed derivations are provided in Appendix A.

### 4.1 Optimization Algorithm

We approximate the posterior distribution of the network weights $\boldsymbol{w}_{1:J}$ using variational inference techniques (Kingma & Welling, 2013; Blei et al., 2017). Specifically, we introduce the *factorized* variational approximation $q(\boldsymbol{w}_{1:J}) = \prod_{j=1}^{J} q(\boldsymbol{w}_j)$, employing an *axis-aligned multivariate* Gaussian distribution for the variational family, i.e. $q(\boldsymbol{w}_j) = \mathcal{N}(\boldsymbol{\mu}_j, \boldsymbol{\rho}_j)$.

---

[2]We note that FedAvg can also be regarded as using a specific first-order algorithm of meta learning called Reptile (Nichol et al., 2018) and setting the outer-loop learning rate to 1.

**Variational Expectation Maximization.** To optimize the marginal likelihood $p(\boldsymbol{y}_{1:J}|\boldsymbol{\omega}, \boldsymbol{\sigma})$, we maximize the evidence lower bound (ELBO):

$$ELBO\left(q(\boldsymbol{w}_{1:J}), \boldsymbol{\omega}, \boldsymbol{\sigma}\right) = \mathbb{E}_{q(\boldsymbol{w}_{1:J})} \log \frac{p(\boldsymbol{y}_{1:J}, \boldsymbol{w}_{1:J}|\boldsymbol{\omega}, \boldsymbol{\sigma})}{q(\boldsymbol{w}_{1:J})}. \tag{5}$$

The ELBO is optimized using Expectation Maximization (EM) (Dempster et al., 1977), a two-stage iterative algorithm involving an E-step and an M-step. Generally, each EM cycle improves the marginal likelihood $p(\boldsymbol{y}_{1:J}|\boldsymbol{\omega}, \boldsymbol{\sigma})$ unless it reaches a local maximum.

**E-step.** At this stage, the latent variables $\{\boldsymbol{\omega}, \boldsymbol{\sigma}\}$ are held fixed. The aim is to maximize ELBO by optimizing the variational approximations $q(\boldsymbol{w}_{1:J})$. By the conditional independence assumption, the objective can be separately optimized. Specifically, each network $j$ minimizes:

$$\mathcal{L}\left(q(\boldsymbol{w}_j)\right) = -\mathbb{E}_{q(\boldsymbol{w}_j)} \log p(\boldsymbol{y}_j|\boldsymbol{w}_j) + D_{KL}(q(\boldsymbol{w}_j)\|p(\boldsymbol{w}_j|\boldsymbol{\omega}, \boldsymbol{\sigma})). \tag{6}$$

The minimization of the negative log-likelihood term is achieved through the minimization of the squared error loss of reconstruction (see Equation (2)). The KL divergence term $D_{KL}$ serves as a regularization constraint on the network weights, pushing the posterior $q(\boldsymbol{w}_j)$ to be closely aligned with the conditional prior determined by $\{\boldsymbol{\omega}, \boldsymbol{\sigma}\}$, which represent the collective mean and variance of all the networks in the ensemble. *The KL divergence term thus serves to couple the neural representations across networks, allowing them to inform each other.*

**M-step.** After obtaining the optimized variational approximations $q(\boldsymbol{w}_{1:J})$, we proceed to maximize the ELBO with respect to the latent variables $\{\boldsymbol{\omega}, \boldsymbol{\sigma}\}$:

$$ELBO(\boldsymbol{\omega}, \boldsymbol{\sigma}) \propto \sum_{j=1}^{J} \mathbb{E}_{q(\boldsymbol{w}_j)} \log p(\boldsymbol{w}_j|\boldsymbol{\omega}, \boldsymbol{\sigma}). \tag{7}$$

Equation (7) allows for a closed-form solution of $\{\boldsymbol{\omega}, \boldsymbol{\sigma}\}$, derived by setting the derivative of the ELBO to zero:

$$\boldsymbol{\omega}^* = \frac{1}{J}\sum_{j=1}^{J}\boldsymbol{\mu}_j, \quad \boldsymbol{\sigma}^* = \frac{1}{J}\sum_{j=1}^{J}\boldsymbol{\rho}_j + (\boldsymbol{\mu}_j - \boldsymbol{\omega}^*)^2. \tag{8}$$

In our framework, $\boldsymbol{\omega}$ serves as the collective mean of individual network weights, while $\boldsymbol{\sigma}$ provides a measure of dispersion. This measure factors in both individual variances and deviations from the collective mean. We note that the KL divergence term, introduced in the preceding E-step objective (see Equation (6)), operates element-wise. *As a result, during the training process, weight elements with larger $\boldsymbol{\sigma}$ values are less regularized, and vice versa. This results in a flexible, self-adjusting regularization scheme that adaptively pushes all weights toward the latent mean, $\boldsymbol{\omega}$.*

### 4.2 Algorithm Implementation

Next, we discuss the intricacies of implementation, addressing in particular the computational challenges associated with Equation (6). See Algorithm 1 for a summary of our method.

**Variational Approximation and Reparameterization Tricks.** Given that the expected likelihood in Equation (6) is generally intractable, we resort to Monte Carlo (MC) sampling to provide an effective estimation. Moreover, we introduce an additional hyperparameter $\beta$ for the KL divergence to balance the trade-off between model complexity and overfitting. Linking the likelihood with the square error loss, for any node $j$, the effective loss function can be expressed as:

$$\mathcal{L}\left(q(\boldsymbol{w}_j)\right) \approx \|\boldsymbol{A}_j(\mathcal{F}_{\widehat{\boldsymbol{w}}_j}(\boldsymbol{C})) - \boldsymbol{y}_j\|_2^2 + \beta D_{KL}(q(\boldsymbol{w}_j)\|p(\boldsymbol{w}_j|\boldsymbol{\omega}, \boldsymbol{\sigma})), \tag{9}$$

where $\widehat{\boldsymbol{w}}_j$ denotes a sample from $q(\boldsymbol{w}_j)$. We only do MC sampling once at each iteration, which works efficiently in practice. To facilitate the gradient-based optimization schemes, we utilize the reparameterization

---

**Algorithm 1** INR-Bayes: Joint reconstruction of INR using Bayesian framework

---

$\quad$ **Input:** $\boldsymbol{\mu}_{1:J}^{(0,0)}, \boldsymbol{\pi}_{1:J}^{(0,0)}, \boldsymbol{\omega}^0, \boldsymbol{\sigma}^0, \eta, \beta, T, R$

$\quad$ **Output:** $\boldsymbol{\mu}_{1:J}^{(R,T)}, \boldsymbol{\pi}_{1:J}^{(R,T)}, \boldsymbol{\omega}^R, \boldsymbol{\sigma}^R$

1: **for** $r = 1$ to $R$ **do**

2: $\quad$ **for** $j = 1, \ldots, J$ **in parallel do**

3: $\quad\quad$ **NodeUpdate**$(\boldsymbol{\omega}^{r-1}, \boldsymbol{\sigma}^{r-1})$

4: $\quad\quad$ After each E-step, collect $\boldsymbol{\mu}_{1:J}^{(r,T)}, \boldsymbol{\pi}_{1:J}^{(r,T)}$.

5: $\quad\quad$ ▷ *Compute the optimal latent variable $\boldsymbol{\omega}, \boldsymbol{\sigma}$.* $\qquad\qquad\qquad\qquad\qquad\qquad\qquad$ ◁

6: $\quad\quad$ $\boldsymbol{\omega}^r = \frac{1}{J} \sum_{j=1}^{J} \boldsymbol{\mu}_j^{(r,T)}$

7: $\quad\quad$ $\boldsymbol{\sigma}^r = \frac{1}{J} \sum_{j=1}^{J} \log\left(1 + \exp(\boldsymbol{\pi}_j^{(r,T)})\right) + (\boldsymbol{\mu}_j^{(r,T)} - \boldsymbol{\omega}^r)^2$

8: **NodeUpdate**$(\boldsymbol{\omega}^r, \boldsymbol{\sigma}^r)$:

9: **for** $t = 1, \ldots, T$ **in parallel do**

10: $\quad$ ▷ *Sample $\widehat{\boldsymbol{w}}_j$.* $\qquad\qquad\qquad\qquad\qquad\qquad\qquad\qquad\qquad\qquad\qquad\qquad\qquad\qquad\qquad$ ◁

11: $\quad$ $\widehat{\boldsymbol{w}}_j^t \sim \boldsymbol{\mu}_j^{(r,t)} + \log\left(1 + \exp(\boldsymbol{\pi}_j^{(r,t)})\right) \mathcal{N}(\boldsymbol{0}, \boldsymbol{I})$

12: $\quad$ ▷ *Compute the loss function.* $\qquad\qquad\qquad\qquad\qquad\qquad\qquad\qquad\qquad\qquad\qquad\qquad$ ◁

13: $\quad$ $\mathcal{L}(\boldsymbol{\mu}_j, \boldsymbol{\rho}_j) = \|\boldsymbol{A}_j(\mathcal{F}_{\widehat{\boldsymbol{w}}_j^t}(\boldsymbol{C})) - \boldsymbol{y}_j\|_2^2 + \beta D_{KL}(q(\boldsymbol{w}_j)\|p(\boldsymbol{w}_j|\boldsymbol{\omega}^r, \boldsymbol{\sigma}^r))$

14: $\quad$ ▷ *SGD on $\boldsymbol{\mu}_j, \boldsymbol{\rho}_j$ with learning rate $\eta$.* $\qquad\qquad\qquad\qquad\qquad\qquad\qquad\qquad\qquad\qquad$ ◁

15: $\quad$ $\boldsymbol{\mu}_j^{(r,t+1)} = \boldsymbol{\mu}_j^{(r,t)} - \eta\frac{\partial\mathcal{L}}{\partial\boldsymbol{\mu}_j}, \boldsymbol{\pi}_j^{(r,t+1)} = \boldsymbol{\pi}_j^{(r,t)} - \eta\frac{\partial\mathcal{L}}{\partial\boldsymbol{\pi}_j}$

---

trick (Kingma & Welling, 2013): $q(\boldsymbol{w}_j) = \boldsymbol{\mu}_j + \log\left(1 + \exp(\boldsymbol{\pi}_j)\right)\mathcal{N}(\boldsymbol{0}, \boldsymbol{I})$. Here, we additionally employ the softplus function in parameterizing the variance $\rho_j$ with the variable $\boldsymbol{\pi}_j$ to ensure the non-negativity of the variance.

The EM algorithm operates by alternating between E and M steps. In the E-step, we perform $T$ iterations to achieve locally optimal variational approximations. Following this, the M-step utilizes the closed-form solution (see Equation (8)) for efficient parameter updating. The entire cycle is executed for $R$ rounds to ensure convergence. Finally, the posterior means $\boldsymbol{\mu}_{1:J}$ serve as the weights for individual neural representations, while the prior mean $\boldsymbol{\omega}$ is used as the weights for the latent neural representation.

## 5 Experiments

**Dataset.** Our study utilizes four CT datasets: WalnutCT with walnut scans (Der Sarkissian et al., 2019), AluminumCT of aluminum alloy at different fatigue-corrosion phases from Tomobank (De Carlo et al., 2018), LungCT from the Medical Segmentation Decathlon (Antonelli et al., 2022) and 4DCT on the lung area (Castillo et al., 2009). Additionally, we include a natural image dataset CelebA (Liu et al., 2015) to evaluate the generalizability to broader applications. The objects of interest are centrally positioned within the images across all datasets. Central positioning in practice is achievable during acquisition by monitoring initial placement.

**Comparison Methods.** We compare our approach with other methods that *do not require* a set of *densely measured data*, thereby excluding those of *supervised learning*: i) Classical techniques: Filtered Back Projection (FBP) and Simultaneous Iterative Reconstruction Technique (SIRT) (Gilbert, 1972); ii) Regularization-based approaches with model-based reconstruction method FISTA (Beck & Teboulle, 2009): total generalized variation denoted as RegTGV proposed by Niu et al. (2014), wavelet based regularization RegWavelet (Zhang et al., 2018a; Garduño et al., 2011) and second order Lysaker-Lundervold-Tai with Rudin-Osher-Fatemi TV, denoted as RegLLT-TV (Kazantsev et al., 2017); iii) Naive INR-based single reconstruction method, denoted as SingleINR; iv) FedAvg, a federated averaging approach proposed by Kundu et al. (2022); v) MAML, a meta-learning technique as discussed by Tancik et al. (2021); vi) INRWild, a method adapted from NeRFWild (Martin-Brualla et al., 2021). FBP and SIRT are classical methods that do not use networks, while all other methods employ an identical INR network as described in the next paragraph.

**INR Network Configuration.** The same backbone and associated configurations are applied to ensure a fair comparison. All INR-based methods employ the SIREN architecture (Sitzmann et al., 2020) coupled with the same positional embedding (Tancik et al., 2020). Specifically, we consistently use a Siren network with the following dimensions: a depth of 8 and width of 128 for the WalnutCT and AluminiumCT experiments, and a depth of 8 with width 256 for the rest of the experiments. Position encoding is implemented using Fourier feature embedding (Tancik et al., 2020), with an embedding dimension of 256 for WalnutCT and AluminiumCT, 512 for the rest. Consistently, this embedded position is input to the INRs to facilitate density prediction. In alignment with Martin-Brualla et al. (2021), `INRWild` utilizes an 8-layer SIREN network for the static segment and a 4-layer SIREN for each transient component. We note that all nodes use the same size network for our method and comparison methods.

**CT Configuration.** We simulate CT projections using the Tomosipo package (Hendriksen et al., 2021) with a parallel beam. Experiments on 4DCT, WalnutCT and CelebA use projections from 40 angles across 180°, while LungCT uses 60 angles and AluminiumCT uses 100 angles. Practical applicability is further tested under a 3D cone-beam CT setting, detailed in Appendix D.1.

**Experiment Configurations.** i) Intra-object: 10 equidistant slices from an object's center. ii) Inter-object: 10 slices from different objects, each from a similar vertical position. iii) 4DCT: 10 temporal phases from one 4DCT slice. All INR-based methods undergo 30K iterations, with `MAML` using the first 10K and `FedAvg` the first 20K for meta initialization, then proceeding to adaptation. For classical methods, `FBP_CUDA` computes the FBP reconstruction while `SIRT_CUDA` from the Astra-Toolbox is used for SIRT, set to run for 5,000 iterations. Hyperparameters like learning rates are optimized using a subset of data (details in Appendix B.4).

**Metrics.** We primarily evaluate using Peak Signal to Noise Ratio (PSNR) and Structural Similarity Index Measure (SSIM), with metrics referenced against ground-truth images. We calculate mean and standard error over all reconstructed images in each experiment.

## 5.1 Results

| Method | Metrics | Inter-walnut | 4DCT | Intra-lung | Inter-Aluminium | Inter-Lung | Inter-Faces* |
|---|---|---|---|---|---|---|---|
| FBP | PSNR | $19.54_{\pm0.25}$ | $25.19_{\pm0.03}$ | $26.50_{\pm0.06}$ | $32.13_{\pm0.05}$ | $24.97_{\pm0.14}$ | $18.12_{\pm0.39}$ |
| | SSIM | $0.328_{\pm0.003}$ | $0.529_{\pm0.001}$ | $0.568_{\pm0.002}$ | $0.717_{\pm0.002}$ | $0.503_{\pm0.004}$ | $0.549_{\pm0.006}$ |
| SIRT | PSNR | $23.82_{\pm0.25}$ | $28.61_{\pm0.03}$ | $28.81_{\pm0.06}$ | $28.61_{\pm0.33}$ | $28.32_{\pm0.13}$ | $29.13_{\pm0.20}$ |
| | SSIM | $0.435_{\pm0.007}$ | $0.746_{\pm0.001}$ | $0.719_{\pm0.002}$ | $0.836_{\pm0.001}$ | $0.678_{\pm0.005}$ | $0.766_{\pm0.005}$ |
| RegTGV | PSNR | $30.64_{\pm0.33}$ | $31.50_{\pm0.04}$ | $31.46_{\pm0.07}$ | $27.21_{\pm0.03}$ | $31.18_{\pm0.19}$ | $28.39_{\pm0.18}$ |
| | SSIM | $0.872_{\pm0.005}$ | $0.859_{\pm0.001}$ | $0.810_{\pm0.002}$ | $0.868_{\pm0.001}$ | $0.809_{\pm0.006}$ | $0.833_{\pm0.005}$ |
| RegWavelet | PSNR | $24.79_{\pm0.25}$ | $27.00_{\pm0.03}$ | $27.95_{\pm0.05}$ | $25.75_{\pm0.03}$ | $27.76_{\pm0.11}$ | $27.00_{\pm0.21}$ |
| | SSIM | $0.460_{\pm0.006}$ | $0.594_{\pm0.001}$ | $0.634_{\pm0.002}$ | $0.817_{\pm0.001}$ | $0.626_{\pm0.004}$ | $0.701_{\pm0.006}$ |
| RegLLT-TV | PSNR | $33.33_{\pm0.30}$ | $32.39_{\pm0.04}$ | $32.26_{\pm0.08}$ | $27.19_{\pm0.03}$ | $31.95_{\pm0.19}$ | $30.09_{\pm0.22}$ |
| | SSIM | $0.933_{\pm0.002}$ | $0.890_{\pm0.001}$ | $0.828_{\pm0.002}$ | $0.867_{\pm0.001}$ | $0.828_{\pm0.006}$ | $\mathbf{0.858}_{\pm0.005}$ |
| SingleINR | PSNR | $35.18_{\pm0.43}$ | $34.07_{\pm0.04}$ | $32.80_{\pm0.11}$ | $35.62_{\pm0.06}$ | $32.64_{\pm0.27}$ | $30.90_{\pm0.32}$ |
| | SSIM | $0.934_{\pm0.004}$ | $0.877_{\pm0.001}$ | $0.815_{\pm0.002}$ | $0.850_{\pm0.001}$ | $0.821_{\pm0.007}$ | $0.831_{\pm0.008}$ |
| INRWild | PSNR | $19.76_{\pm0.35}$ | $33.76_{\pm0.04}$ | $28.46_{\pm0.07}$ | $29.77_{\pm0.05}$ | $25.05_{\pm0.18}$ | $16.43_{\pm0.28}$ |
| | SSIM | $0.675_{\pm0.010}$ | $0.863_{\pm0.001}$ | $0.674_{\pm0.003}$ | $0.694_{\pm0.002}$ | $0.560_{\pm0.08}$ | $0.260_{\pm0.011}$ |
| MAML | PSNR | $35.66_{\pm0.38}$ | $34.35_{\pm0.05}$ | $33.26_{\pm0.10}$ | $36.22_{\pm0.06}$ | $33.13_{\pm0.22}$ | $25.71_{\pm0.27}$ |
| | SSIM | $0.941_{\pm0.003}$ | $0.881_{\pm0.001}$ | $0.825_{\pm0.002}$ | $0.871_{\pm0.001}$ | $0.833_{\pm0.006}$ | $0.647_{\pm0.012}$ |
| FedAvg | PSNR | $31.57_{\pm0.32}$ | $34.67_{\pm0.04}$ | $32.42_{\pm0.08}$ | $35.75_{\pm0.06}$ | $31.68_{\pm0.19}$ | $30.74_{\pm0.29}$ |
| | SSIM | $0.886_{\pm0.004}$ | $0.885_{\pm0.001}$ | $0.808_{\pm0.002}$ | $0.849_{\pm0.001}$ | $0.807_{\pm0.006}$ | $0.827_{\pm0.007}$ |
| INR-Bayes | PSNR | $\mathbf{36.13}_{\pm0.33}$ | $\mathbf{34.79}_{\pm0.04}$ | $\mathbf{33.90}_{\pm0.10}$ | $\mathbf{36.67}_{\pm0.06}$ | $\mathbf{33.75}_{\pm0.20}$ | $\mathbf{31.31}_{\pm0.31}$ |
| | SSIM | $\mathbf{0.946}_{\pm0.003}$ | $\mathbf{0.894}_{\pm0.001}$ | $\mathbf{0.840}_{\pm0.002}$ | $\mathbf{0.881}_{\pm0.001}$ | $\mathbf{0.847}_{\pm0.006}$ | $0.847_{\pm0.007}$ |

Table 1: Performance comparison across different settings on noiseless measurements. The highest average PSNR/SSIM values are highlighted in bold, consistently showing that INR-Bayes outperforms other methods. *Reconstruction of human faces is included to illustrate our method's broad applicability, despite lacking practical relevance in physical contexts.

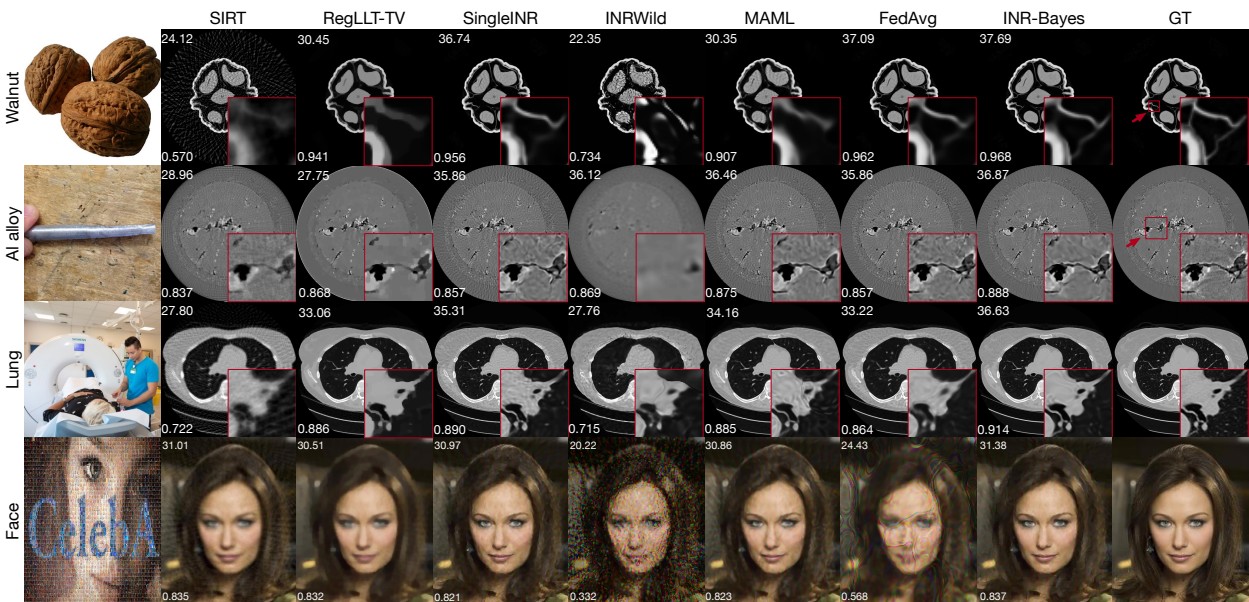

Figure 3: Visual comparison of reconstruction performance on noiseless measurements. Enlarged areas are highlighted in red insets. PSNR values are on the top left, with SSIM values on the bottom left. Reconstruction of human faces is included to illustrate our method's broad applicability, despite lacking practical relevance in physical contexts. Illustrations figures (first column) are modified from J.Dncsn (2009); Agashi5859 (2014); Lene (2021); Liu et al. (2015).

**Reconstruction Performance.** Table 1 shows that our method consistently achieves the highest average metrics across datasets on noiseless measurements, with the exception of the SSIM values on the CelebA dataset, excelling in leveraging inherent trends in slices. This advantage is particularly pronounced in experiments with distinct transition patterns, as seen in both inter-object and intra-object settings. In contrast, `INRWild` generally underperforms compared to `SingleINR`. Its strategy of jointly reconstructing static representations while independently capturing transient characteristics fails to enhance overall reconstruction quality. In most scenarios, `FedAvg` is outperformed by `SingleINR`, suggesting its inability to learn a meta representation that meaningfully improves reconstruction quality. `MAML` ranks as the second-best method in most settings, indicating its learned meta representation aids in individual reconstruction performance. However, regularization-based methods typically underperform compared to INR-based methods, except on the CelebA dataset where `RegLLT-TV` achieves the highest SSIM value. This underscores the efficacy of the heuristic LLT-TV prior for natural images as opposed to CT images.

The visual comparisons in Figure 3 further validate our results. Reconstructions by `SingleINR` exhibit noticeable artifacts, while `FedAvg` and `MAML` struggle to capture finer image details. `RegLLT-TV` often results in overly smooth reconstructions, but in contrast, `INR-Bayes` achieves superior visual quality, effectively balancing smoothness and detail preservation, yielding results closer to the ground truth.

**Overfitting.**[3] Iterative reconstruction methods tend to overfit when applied to limited data (Herman & Odhner, 1991; Effland et al., 2020) (particularly noticeable in sparse-view CT scenarios). In contrast, Bayesian frameworks have demonstrated robustness against overfitting (MacKay, 1992; Neal, 2012; Blundell et al., 2015). To validate this, we extend the training iterations in Table 1 from 30K to 60K. As shown in Figure 4 on inter-object LungCT, the learning curves of baselines deteriorate in the long run, indicating overfitting

---

[3]In the context of CT reconstruction, conventional iterative methods like SIRT typically approach a good approximation to the exact solution early in iterations and subsequently diverge from it, often attributed to convergence issues (Elfving et al., 2014). In contrast, in the machine learning domain, overfitting refers to a scenario where a neural network excessively fits to training data, leading to a drop in actual performance despite decreasing training loss (Srivastava et al., 2014). Our experiments contain both non-learning-based methods (SIRT) and learning-based methods (INR). We acknowledge the subtle differences between convergence challenges and overfitting but use the term `overfitting` broadly to describe performance deterioration in both noiseless and noisy scenarios.

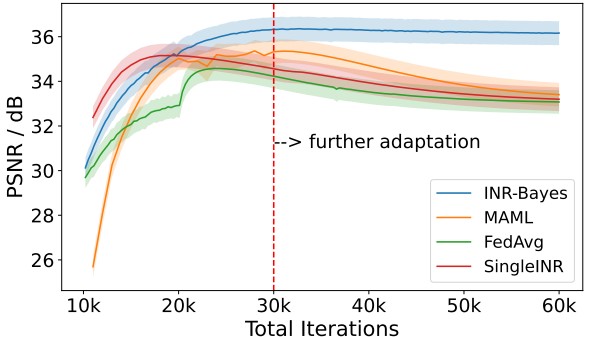
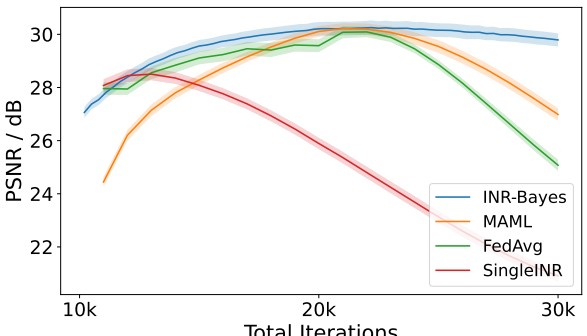

Figure 4: After the initial 30,000 iterations, each joint reconstruction method undergoes an additional 30,000 iterations for further adaptation.

Figure 5: The training curve of different methods on noisy measurement. SingleINR, MAML, and FedAvg overfit strongly in 30,000 iterations.

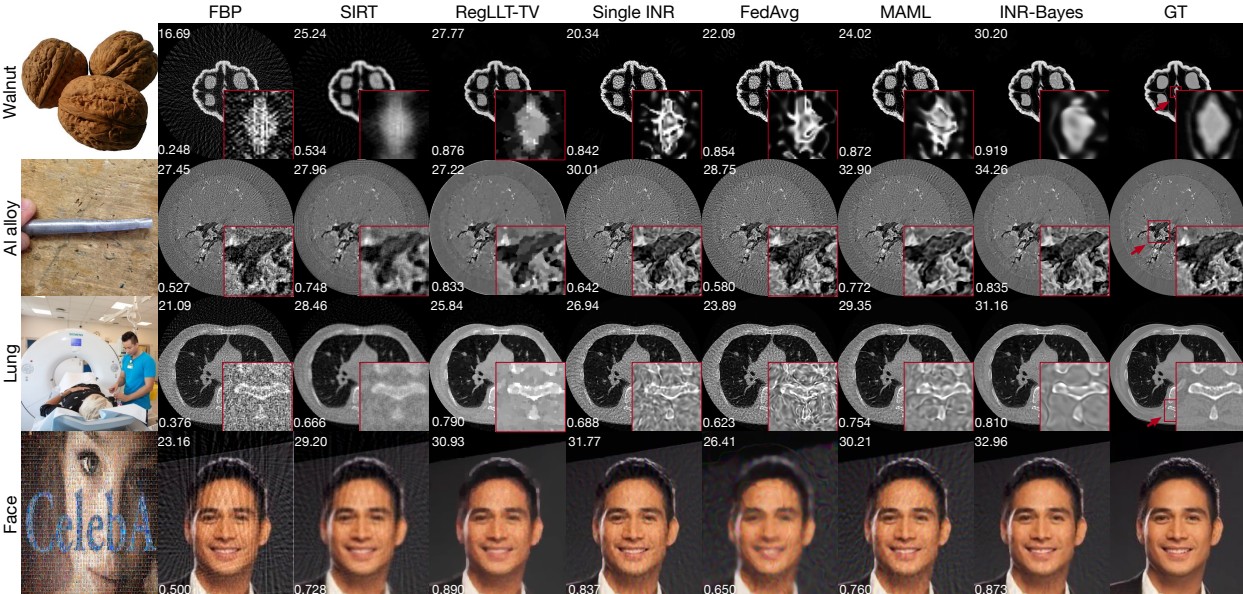

Figure 6: Performance comparison on inter-object noisy measurements. Enlarged areas are shown in red insets. PSNR and SSIM values are provided in the top left and bottom left corners, respectively. Illustrations figures (first column) are modified from J.Dncsn (2009); Agashi5859 (2014); Lene (2021); Liu et al. (2015).

during optimization. Conversely, our approach maintains a consistent level of reconstruction quality once the optimal performance is achieved, underscoring its robustness.

Given that practical measurements often contain noise, leading to poorer reconstruction quality, we evaluate the methods on noisy measurements on AluminumCT, WalnutCT, LungCT and CelebA datasets in inter-object setup. The noise is simulated following Hendriksen et al. (2020) (detailed in Appendix B.1). As indicated in Table 2, our method's performance is notably superior against other methods under noise conditions. Figure 5 shows the training curves on noisy WalnutCT data. Although all methods reach similar peak PSNR values, the compared baselines rapidly deteriorate due to overfitting, while our method exhibits minimal overfitting thanks to its continuous prior guidance. The reconstruction outcomes for these noisy scenarios are depicted in Figure 6. This robustness is especially critical in practical scenarios, where determining an exact stopping criterion is challenging, as the ground truth reference is not available.

**Challenge in Determining Appropriate Stopping Criterion.** We find that in the noisy CT reconstruction environments it is inherently challenging to determine the optimal stopping point when the overfitting problem presents. This is due to the variation of convergence speed across different objects. To demonstrate that, we present the individual SingleINR training curves from three different patients within

| Method | Noisy Walnut | | Noisy Lung | | Noisy Alumnium | | Noisy Faces | |
|---|---|---|---|---|---|---|---|---|
| | PSNR | SSIM | PSNR | SSIM | PSNR | SSIM | PSNR | SSIM |
| FBP | 15.59 $\pm 0.26$ | 0.270 $\pm 0.002$ | 20.33 $\pm 0.17$ | 0.353 $\pm 0.003$ | 27.77 $\pm 0.04$ | 0.515 $\pm 0.001$ | 17.81 $\pm 0.38$ | 0.474 $\pm 0.007$ |
| SIRT | 23.48 $\pm 0.25$ | 0.405 $\pm 0.007$ | 28.03 $\pm 0.14$ | 0.623 $\pm 0.005$ | 28.23 $\pm 0.03$ | 0.759 $\pm 0.001$ | 26.90 $\pm 0.21$ | 0.681 $\pm 0.007$ |
| RegTGV | 27.76 $\pm 0.31$ | 0.748 $\pm 0.010$ | 27.44 $\pm 0.17$ | 0.619 $\pm 0.006$ | 27.14 $\pm 0.03$ | 0.862 $\pm 0.001$ | 27.97 $\pm 0.18$ | 0.800 $\pm 0.004$ |
| RegWavelet | 23.49 $\pm 0.25$ | 0.393 $\pm 0.006$ | 26.52 $\pm 0.15$ | 0.548 $\pm 0.005$ | 25.19 $\pm 0.03$ | 0.716 $\pm 0.001$ | 22.83 $\pm 0.26$ | 0.471 $\pm 0.010$ |
| RegLLT-TV | 28.51 $\pm 0.31$ | 0.817 $\pm 0.010$ | 30.58 $\pm 0.17$ | 0.783 $\pm 0.006$ | 27.23 $\pm 0.03$ | 0.863 $\pm 0.001$ | 28.00 $\pm 0.18$ | **0.801** $\pm 0.001$ |
| SingleINR | 22.03 $\pm 0.39$ | 0.809 $\pm 0.006$ | 27.16 $\pm 0.23$ | 0.681 $\pm 0.008$ | 29.50 $\pm 0.12$ | 0.602 $\pm 0.006$ | 27.41 $\pm 0.24$ | 0.701 $\pm 0.009$ |
| MAML | 26.55 $\pm 0.30$ | 0.860 $\pm 0.004$ | 30.04 $\pm 0.20$ | 0.758 $\pm 0.006$ | 32.58 $\pm 0.07$ | 0.741 $\pm 0.003$ | 27.55 $\pm 0.26$ | 0.701 $\pm 0.010$ |
| FedAvg | 25.37 $\pm 0.39$ | 0.836 $\pm 0.005$ | 27.13 $\pm 0.22$ | 0.696 $\pm 0.006$ | 29.87 $\pm 0.10$ | 0.617 $\pm 0.005$ | 25.67 $\pm 0.15$ | 0.669 $\pm 0.007$ |
| INR-Bayes | **30.13** $\pm 0.31$ | **0.885** $\pm 0.003$ | **31.20** $\pm 0.20$ | **0.799** $\pm 0.006$ | **35.09** $\pm 0.12$ | **0.869** $\pm 0.006$ | **28.24** $\pm 0.27$ | 0.729 $\pm 0.010$ |

Table 2: Comparison of reconstruction performance on noisy measurements on AluminumCT, WalnutCT, LungCT and CelebA dataset.

the noisy LungCT dataset in Figure 7. Despite originating from the same dataset, these three cases exhibit distinctly different optimal stopping points, suggesting that a one-size-fits-all approach to training termination is ineffective for appraches suffering from the overfitting problem.

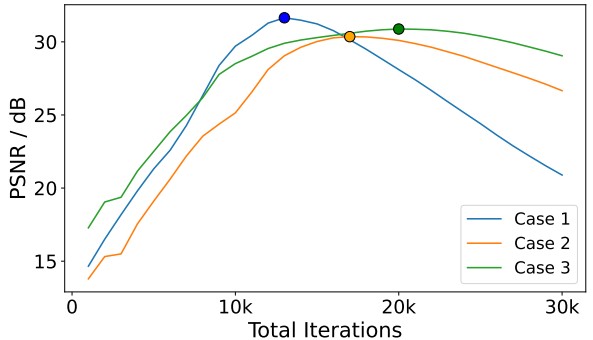

Figure 7: SingleINR training curves for three different patients from the noisy LungCT dataset, each showing unique optimal stopping points. This variability exemplifies the challenges in universally determining the precise moment to stop training.

**Applying to Unseen Data using Learned Prior.** To evaluate the generalizability of the learned priors, we apply meta models or latent variables from the inter-object LungCT experiment to guide the reconstruction of new subjects in the dataset. Specifically, we select 5 consecutive slices from new objects, choosing slices from a similar location (i.e. the similar axial slice index) the prior has been trained. The prior information is solely utilized to guide the reconstruction and is not updated during the process. Table 3 shows that `FedAvg` fails to improve the reconstruction quality compared with `singleINR`, suggesting its learned meta neural representation struggles to generalize to unseen data. In contrast, both `INR-Bayes` and `MAML` effectively leverage their trained priors for improved reconstruction, with our method showing notably better metrics. Applying learned prior can also accelerate the training process. Figure 8 presents learning curves of different methods. All joint reconstruction methods converge faster than individual reconstruction. Initially, `FedAvg` converges the fastest, but as training progresses, both `MAML` and `INR-Bayes` surpass it. Additionally, the results reconfirm the robustness of `INR-Bayes` against overfitting, an issue that tends to challenge other compared methods.

| | SingleINR | FedAvg | MAML | INR-Bayes |
|---|---|---|---|---|
| PSNR | 30.22 $\pm 0.12$ | 29.89 $\pm 0.16$ | 30.42 $\pm 0.14$ | **31.31** $\pm 0.12$ |
| SSIM | 0.769 $\pm 0.003$ | 0.754 $\pm 0.004$ | 0.770 $\pm 0.003$ | **0.799** $\pm 0.003$ |

Table 3: Adaptation on new patients using priors learned from other patients compared to individual reconstruction. The results are averaged by 10 new patients (5 slices each patient).

**Comparison with Different Numbers of Angles.** Next, we investigate how different methods fare with varying levels of data sparsity. Figure 9 demonstrates that all methods benefit from increased scanning angles. Methods that leverage prior information like `FedAvg`, `MAML`, and `INR-Bayes` outperform `singleINR` when the number of angles is limited. With only 20 angles, `FedAvg`'s performance is on par with our method, indicating that the averaging scheme can be effective in extremely data-scarce scenarios. However, as the number

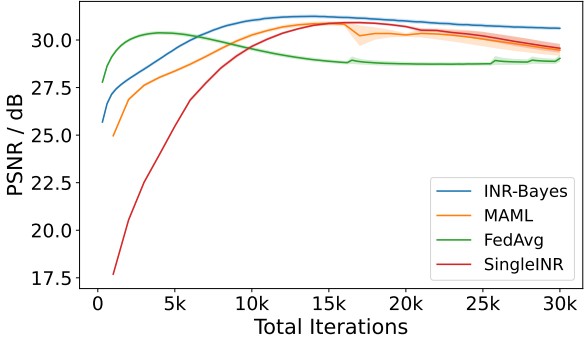

Figure 8: Reconstruction of 5 new patients using the learned prior from 10 other patients, compared to individual reconstruction. All joint methods reconstruct faster with the priors they learned.

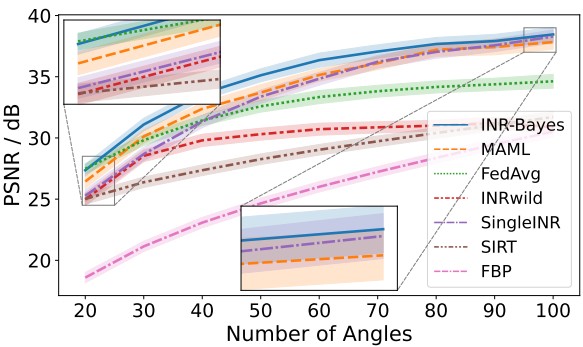

Figure 9: Performance across different numbers of scanning angles. Our method shows an advantage over other comparison methods, especially in sparse angle cases.

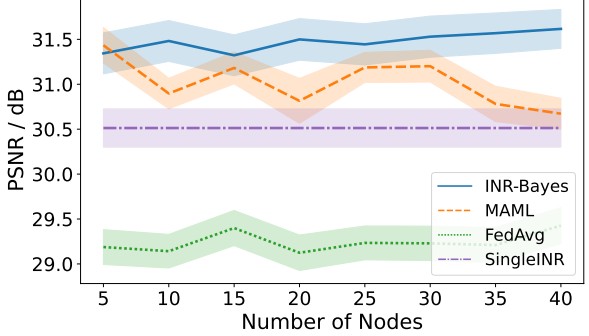

Figure 10: The performance of different methods on new patients using prior obtained with different numbers of patients. SingleINR is presented as a reference. The performance of our propsoed method gradually increases with the increasing number joint reconstruction nodes.

of angles grows, both `INR-Bayes` and `MAML` surpass `FedAvg`. It is also worth noting that the performance gap between `singleINR` and `INR-Bayes` narrows as more data becomes available, suggesting that while the prior information is useful in sparse data situations, its advantage diminishes in the data-rich environment. Nevertheless, our `INR-Bayes` method generally yields the best results in various settings.

**Impact of Joint Nodes Numbers on Learned Prior.** We explore how varying the number of nodes influences the learned prior in an inter-object LungCT dataset setup. Figure 10 illustrates that our `INR-Bayes` method benefits from an increased number of nodes, unlike `FedAvg`, which consistently lags behind in performance. Notably, `MAML` shows a decline in performance with more nodes, consistent with the trends observed in Figure 17. These findings suggest that `INR-Bayes` effectively develops a stronger prior with more nodes. As the number of nodes increases, the prior's mean and variance gradually align with population statistics, thereby enhancing its utility for new object reconstructions. This advantage is expected to continue until reaching a saturation point with a sufficiently large number of nodes.

**Computation.** Our method takes 22.7 minutes on average for a $501 \times 501$ reconstruction on A100 GPU, 15% longer than other INR methods. This computational overhead yields noticeable gains in reconstruction quality and robustness across diverse settings. We also investigate the computational efficiency and performance trade-off in Appendix D.7.

# 6 Conclusions and Limitations

In this work, we study the possibility of improving the CT reconstruction quality through joint reconstruction and introduce a novel INR-based Bayesian framework. Through extensive experiments, our method has effectively showcased its ability to leverage the statistical regularities inherent in the sparse and noisy measurements of multiple objects to improve individual reconstructions. This capability allows our approach to outperform compared methods in reconstruction quality, robustness to overfitting as well as generalizability.

While the primary focus of our method has been on joint CT reconstruction, its underlying principles hold potential applicability across a variety of inverse problems that are challenged by sparse measurements.

We recognize that INR-based methods outperform conventional ones but require more computation, making their efficiency a crucial focus for future research. Additionally, the metrics employed in our study may not always correlate with clinical evaluations (Renieblas et al., 2017; Verdun et al., 2015). If applied in medical fields, clinical verification of our method remains essential to understand its practical implications and efficacy.

### Acknowledgments

Jiayang Shi, Daniël M. Pelt, K. Joost Batenburg and Matthew B. Blaschko are supported by the European Union H2020-MSCA-ITN-2020 under grant agreement no. 956172 (xCTing). Matthew B. Blaschko and Junyi Zhu received funding from the Flemish Government (AI Research Program) and the Research Foundation - Flanders (FWO) through project number G0G2921N.

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

## A    Appendix: Variational Expectation Maximization of Marginal Likelihood

To optimize the marginal likelihood $p(\boldsymbol{y}_{1:J}|\boldsymbol{\omega}, \boldsymbol{\sigma})$, we leverage a variational approximation of the network weights $q(\boldsymbol{w}_{1:J})$. We start by rewriting the marginal likelihood as follows:

$$\log p(\boldsymbol{y}_{1:J}|\boldsymbol{\omega}, \boldsymbol{\sigma}) = \mathbb{E}_{q(\boldsymbol{w}_{1:J})} \log \frac{p(\boldsymbol{y}_{1:J}, \boldsymbol{w}_{1:J}|\boldsymbol{\omega}, \boldsymbol{\sigma}) q(\boldsymbol{w}_{1:J})}{q(\boldsymbol{w}_{1:J}) p(\boldsymbol{w}_{1:J}|\boldsymbol{y}_{1:J}, \boldsymbol{\omega}, \boldsymbol{\sigma})} \tag{10}$$

$$= \underbrace{\mathbb{E}_{q(\boldsymbol{w}_{1:J})} \log \frac{p(\boldsymbol{y}_{1:J}, \boldsymbol{w}_{1:J}|\boldsymbol{\omega}, \boldsymbol{\sigma})}{q(\boldsymbol{w}_{1:J})}}_{ELBO(q(\boldsymbol{w}_{1:J}), \boldsymbol{\omega}, \boldsymbol{\sigma})} + D_{KL}\left(q(\boldsymbol{w}_{1:J}) \| p(\boldsymbol{w}_{1:J}|\boldsymbol{y}_{1:J}, \boldsymbol{\omega}, \boldsymbol{\sigma}))\right). \tag{11}$$

Since directly maximizing the marginal likelihood is computationally infeasible, we instead maximize its variational lower bound, commonly known as the evidence lower bound (ELBO):

$$ELBO(q(\boldsymbol{w}_{1:J}), \boldsymbol{\omega}, \boldsymbol{\sigma}) = \mathbb{E}_{q(\boldsymbol{w}_{1:J})} \log \frac{p(\boldsymbol{y}_{1:J}, \boldsymbol{w}_{1:J}|\boldsymbol{\omega}, \boldsymbol{\sigma})}{q(\boldsymbol{w}_{1:J})} \tag{12}$$

$$= \mathbb{E}_{q(\boldsymbol{w}_{1:J})} \log p(\boldsymbol{y}_{1:J}|\boldsymbol{w}_{1:J}, \boldsymbol{\omega}, \boldsymbol{\sigma}) - D_{KL}(q(\boldsymbol{w}_{1:J}) \| p(\boldsymbol{w}_{1:J}|\boldsymbol{\omega}, \boldsymbol{\sigma})). \tag{13}$$

The measurements are mutually independent and do not depend on the latent variables $\{\boldsymbol{\omega}, \boldsymbol{\sigma}\}$ given network weights $\boldsymbol{w}_{1:J}$. Moreover, each network is trained solely on its corresponding measurement. Therefore, the log likelihood decomposes into: $\mathbb{E}_{q(\boldsymbol{w}_{1:J})} \log p(\boldsymbol{y}_{1:J}|\boldsymbol{w}_{1:J}, \boldsymbol{\omega}, \boldsymbol{\sigma}) = \sum_{j=1}^{J} \mathbb{E}_{q(\boldsymbol{w}_j)} \log p(\boldsymbol{y}_j|\boldsymbol{w}_j)$. Additionally, due to the factorized variational approximation $q(\boldsymbol{w}_{1:J}) = \prod_{j=1}^{J} q(\boldsymbol{w}_j)$ and the assumption of conditional independence $p(\boldsymbol{w}_{1:J}|\boldsymbol{\omega}, \boldsymbol{\sigma}) = \prod_{j=1}^{J} p(\boldsymbol{w}_j|\boldsymbol{\omega}, \boldsymbol{\sigma})$, we can obtain the following form of ELBO:

$$ELBO(q(\boldsymbol{w}_{1:J}), \boldsymbol{\omega}, \boldsymbol{\sigma}) = \sum_{j=1}^{J} \mathbb{E}_{q(\boldsymbol{w}_j)} \log p(\boldsymbol{y}_j|\boldsymbol{w}_j) - D_{KL}(q(\boldsymbol{w}_j) \| p(\boldsymbol{w}_j|\boldsymbol{\omega}, \boldsymbol{\sigma})). \tag{14}$$

Since each network is trained separately, our framework runs efficiently via parallel computing. The likelihood $p(\boldsymbol{y}_j|\boldsymbol{w}_j)$ can be maximized by minimizing the squared error loss of the reconstruction (see Equation (2)). Similarly to Zhu et al. (2023), we adopt EM to maximize the ELBO. At the E-step, the latent variables $\{\boldsymbol{\omega}, \boldsymbol{\sigma}\}$ are fixed and each network optimizes with respect to the loss function:

$$\mathcal{L}(q(\boldsymbol{w}_j)) = \mathbb{E}_{q(\boldsymbol{w}_j)}\|\boldsymbol{A}\mathcal{F}_{\boldsymbol{w}_j}(\boldsymbol{C}) - \boldsymbol{y}_j\|_2^2 + D_{KL}(q(\boldsymbol{w}_j)\|p(\boldsymbol{w}_j|\boldsymbol{\omega}, \boldsymbol{\sigma})). \tag{15}$$

The expectation can be estimated via MC sampling. When the variational approximations $q(\boldsymbol{w}_j) \sim \mathcal{N}(\boldsymbol{\mu}_j, \boldsymbol{\rho}_j)$, $j = 1 \ldots J$ are formed, we can perform the M-step with the $q(\boldsymbol{w}_{1:J})$ being fixed. Simplifying Equation (14):

$$ELBO(\boldsymbol{\omega}, \boldsymbol{\sigma}) = -\sum_{j=1}^{J} D_{KL}\left(q(\boldsymbol{w}_j)\|p(\boldsymbol{w}_j|\boldsymbol{\omega}, \boldsymbol{\sigma})\right) \tag{16}$$

$$\propto \sum_{j=1}^{J} \mathbb{E}_{q(\boldsymbol{w}_j)} \log p(\boldsymbol{w}_j|\boldsymbol{\omega}, \boldsymbol{\sigma}) \tag{17}$$

$$\propto -\sum_{j=1}^{J} \log |\operatorname{diag}(\boldsymbol{\sigma})| + (\boldsymbol{\omega} - \boldsymbol{\mu}_j)^2 \cdot \boldsymbol{\sigma}^{-1} + \boldsymbol{\rho}_j \cdot \boldsymbol{\sigma}^{-1}, \tag{18}$$

where $\operatorname{diag}(\boldsymbol{\sigma})$ represent a diagonal matrix with the diagonal $\boldsymbol{\sigma}$. The above equation delivers a closed-form solution by setting the derivative to zero:

$$\frac{\partial}{\partial \boldsymbol{\omega}} ELBO(\boldsymbol{\omega}, \boldsymbol{\sigma}) \propto \sum_{j=1}^{J} \boldsymbol{\omega} - \boldsymbol{\mu}_j := 0 \quad \Rightarrow \quad \boldsymbol{\omega}^* = \frac{1}{J} \sum_{j=1}^{J} \boldsymbol{\mu}_j. \tag{19}$$

$$\frac{\partial}{\partial \boldsymbol{\sigma}} ELBO(\boldsymbol{\omega}, \boldsymbol{\sigma}) \propto \sum_{j=1}^{J} \boldsymbol{\sigma} - (\boldsymbol{\omega} - \boldsymbol{\mu}_j)^2 - \boldsymbol{\rho}_j := 0 \quad \Rightarrow \quad \boldsymbol{\sigma}^* = \frac{1}{J} \sum_{j=1}^{J} (\boldsymbol{\omega}^* - \boldsymbol{\mu}_j)^2 + \boldsymbol{\rho}_j. \tag{20}$$

# B Appendix: Details for Experiments

## B.1 Noise Simulation for CT Reconstructions

We incorporate noise in our CT simulations by applying Poisson noise to measurements, following the approach described in (Hendriksen et al., 2020). According to Beer-Lambert's law (Beer, 1852), the photon count detected $I^*$ is a function of the initial incoming photon count $I_0$, the absorption factor $\gamma$ of the object, and the measurement $y_0$, expressed as:

$$I^* = I_0 \exp(\gamma y_0). \tag{21}$$

The noisy measurement $y$ is modeled by introducing variability into the detected photon count. This is achieved by simulating the detected intensity $\hat{I}$ as a Poisson-distributed variable relative to the scenario:

$$I_0 \exp(\gamma y) = \hat{I} \sim Poisson(I_0 \exp(\gamma y_0)), \tag{22}$$

$$y = -\gamma^{-1} \log \frac{\hat{I}}{I_0}. \tag{23}$$

In our experiments, we calibrate the noise level using two parameters: the average absorption $\gamma$ and the photon count $I_0$. For the WalnutCT dataset, we set $\gamma$ to 50% and $I_0$ to 5000. For the LungCT dataset, we similarly set $\gamma$ to 50%, but with a higher photon count of $I_0$ at 20000. This differential setting introduces varying degrees of noise to the measurements, as illustrated in Figure 11. The figure showcases how sparse measurements inherently challenge CT reconstruction, resulting in blurred images due to insufficient data. The addition of simulated noise further complicates this challenge by adding noise to the already blurred images from sparse-view reconstructions.

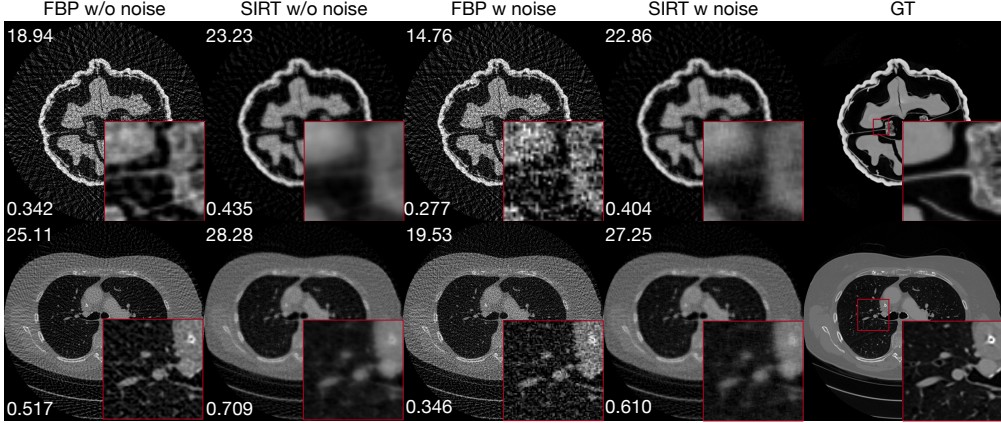

Figure 11: Visual comparison of sparse CT reconstruction with and without simulated noise.

## B.2 INRWild

We implement `INRWild` using the Siren network architecture (Sitzmann et al., 2020). Specifically, the static component, $\mathcal{G}_{\phi}$, is represented using a standard 8-layer Siren network. In contrast, each of the transient parts, $\mathcal{H}_{w_j}$, is characterized by a more compact 4-layer Siren network following the design from `NeRFWild` (Martin-Brualla et al., 2021). Each transient part is linked to a unique one-hot vector, which is embedded into a 16-dimensional transient feature $b_j$. This feature, alongside intermediate features $\mathcal{G}_{\phi}^{\backslash r}(C)$ from the static network, serves as input for transient modeling. The transient feature $b_j$ is optimized during joint training.

The optimization process ensures that the static and transient components are jointly optimized to ensure distinct representations, as articulated in Equation (3). A visual representation of the `INRWild` structure is provided in Figure 12.

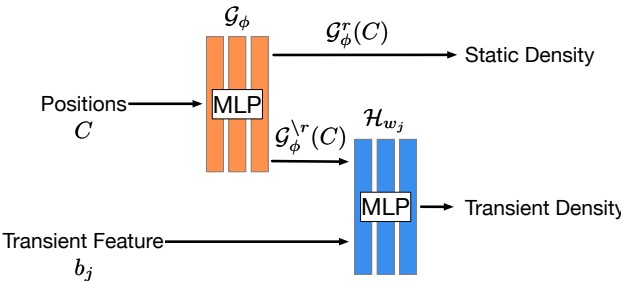

Figure 12: Schematic depiction of INRWild: A tailored version adapted from NeRFWild (Martin-Brualla et al., 2021) for joint CT reconstruction.

## B.3 INR-based Network Configuration

In this study, all INR-based methods employ the Siren network (Sitzmann et al., 2020) as their core architecture. For each experimental setup, we consistently use a Siren network with specific dimensions: a depth of 8 and width of 128 for the WalnutCT experiments, and a depth of 8 with width 256 for the LungCT and 4DCT experiments. Position encoding is implemented using Fourier feature embedding (Tancik et al., 2020), with an embedding dimension of 512 for LungCT and 4DCT, and 256 for WalnutCT. Consistently, this embedded position is input to the INRs to facilitate density prediction.

### B.4 Configuration of All Methods

**Individual Reconstruction Methods.** For 2D experiments, we employ the `FBP` method, and for 3D experiments, its counterpart, FDK. GPU-accelerated operations, `FBP_CUDA` for 2D and `FDK_CUDA` for 3D, are sourced from the Astra-Toolbox (Van Aarle et al., 2016). The iterative method `SIRT`, specifically the `SIRT_CUDA` operation, also from the Astra-Toolbox, is configured with 5,000 iterations. INR-based methods are set to operate for 30,000 iterations. The iteration counts for `SIRT` and INR-based methods are determined through preliminary tests on a dataset subset to ensure optimal performance.

**Hyperparameters.** We tune the hyperparameters of our method and baselines in the inter-object scenario and subsequently apply to all experiments. In particular, for the federated averaging approach `FedAvg` (Kundu et al., 2022), we average individual reconstructions every 100 iterations, amounting to a total of 200 averaging iterations, followed by 10,000 individual adaptation iterations using the learned meta initialization. For the meta-learning technique `MAML` (Tancik et al., 2021), which is rooted in the `MAML` update policy (Finn et al., 2017), we designate a meta learning phase spanning 10,000 iterations (10 inner steps and 1000 outer iterations) across all measurements. This is succeeded by 20,000 individual adaptation iterations utilizing the learned meta initialization. Our `INR-Bayes` undergoes 300 EM loops. For each loop, the E-step iterates 100 times to update the posterior approximation. All INR-based models utilize the Adam optimizer Kingma & Ba (2014) with the first moment 0.9 and the second moment 0.999. The learning rate is set to $1 \times 10^{-5}$. For our method, the additional hyperparameter $\beta$ for the KL divergence term is determined as $1 \times 10^{-14}$ for WalnutCT and $1 \times 10^{-16}$ for LungCT, 4DCT.

**Final Reconstructrion.** After training, ours `INR-Bayes` yields a parameter distribution. Common methods for producing the final result based on this distribution include sampling the network parameters multiple times and averaging the network outputs, or using the mode of the distribution as the network parameters to produce the result (i.e., maximum a posteriori). In our work, we choose the latter approach, as it empirically yields better performance. For the baseline methods, we use the final trained model to produce the final reconstruction.

## C Appendix: Dataset Details

**4DCT.** This dataset Castillo et al. (2009), sized $10 \times 136 \times 512 \times 512$, contains 136 CT image slices captured across 10 respiratory phases of one patient. The main variations across these phases are due to respiratory movements, such as the lungs' expansion and contraction.

**LungCT.** Comprising CT scans from 96 patients Antonelli et al. (2022), its volumes range from $112 \times 512 \times 512$ to $636 \times 512 \times 512$. Despite inherent similarities representing human lungs, individual scan features can vary significantly. Slices within a volume show a consistent pattern, yet fewer stationary features are shared between them. The dataset comprises scans both with and without tumors. For our experiments, we randomly selected patients and images without distinguishing between those containing tumors and those without, aiming for a diverse representation of lung CT images.

**WalnutCT.** This dataset Der Sarkissian et al. (2019) features 42 CT volumes of walnuts, each sized $500 \times 501 \times 501$.

**AluminiumCT.** Comprising 25 CT scans of an aluminium alloy tested across various fatigue cycles De Carlo et al. (2018), each original scan measures $2160 \times 2560 \times 2560$. For computational feasibility, we preprocess these by averaging every five columns in each detector row, reducing the resolution to $2160 \times 512 \times 512$. This dataset allows us to explore the structural integrity and detect minute defects within the material under different stress conditions, simulating practical industrial applications.

**CelebA.** This dataset Li et al. (2021) consists of 202,599 celebrity face images of dimension $3 \times 218 \times 178$.

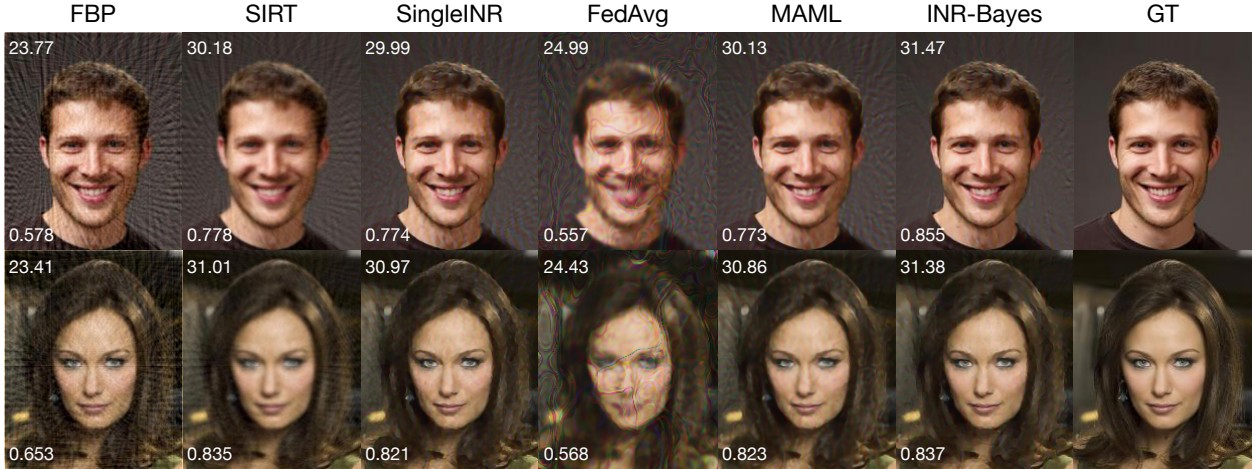

Figure 13: Visual comparison on human faces results. PSNR values are on the top left, with SSIM values on the bottom left.

# D   Appendix: Extra Results

## D.1   3D Joint Reconstruction

To assess the real-world viability of our method, we conduct evaluations in a 3D cone-beam CT context, which more closely aligns with practical scenarios. We choose CT volumes from different patients of size $128^3$, ensuring they represent analogous regions of the human body. The projections are simulated with 40 angles spanning a full $360°$ rotation.

We conduct experiments on 9 groups of joint reconstructions, with each group jointly reconstructing 10 different patients' CT volumes, each sized $128^3$. Table 4 displays the results. Consistent with the findings from 2D CT experiments, our approach surpasses other comparative methods, substantiating its practical relevance. MAML displays slightly inferior performance compared to SingleINR. A potential rationale for this could be that, given the augmented data volume, meta-learning might necessitate extended meta-learning iterations to glean a meaningful representation.

|  | FDK | SIRT | SingleINR | FedAvg | MAML | INR-Bayes |
|---|---|---|---|---|---|---|
| PSNR | 19.40 $\pm 0.62$ | 24.91 $\pm 0.45$ | 33.99 $\pm 0.42$ | 30.30 $\pm 0.27$ | 33.67 $\pm 0.55$ | **34.19** $\pm 0.38$ |
| SSIM | 0.550 $\pm 0.005$ | 0.650 $\pm 0.007$ | 0.932 $\pm 0.007$ | 0.862 $\pm 0.004$ | 0.932 $\pm 0.009$ | **0.945** $\pm 0.004$ |

Table 4: Results from 3D cone-beam CT reconstruction. The highest average PSNR/SSIM values that are statistically significant are highlighted in bold.

## D.2   Joint Reconstruction on Human Faces

To offer a deeper insight, we visualize the learned priors across different joint reconstruction techniques. Figure 14 illustrates INRWild's extraction of "static" and "transient" components from varying faces. Notably, INRWild captures a generalized "face"-like static component. However, due to the significant disparities among face images, this generalized extraction does not significantly enhance individual reconstructions. Figure 15 showcases the learned priors from FedAvg, MAML, and our approach INR-Bayes. Both FedAvg and INR-Bayes succeed in deriving an interpretable prior. However, the face-like meta initialization does not directly benefit reconstruction in the case of FedAvg. In contrast, MAML struggles to capture a face-like prior during its preliminary phase, but obtains final reconstructions better than FedAvg. Our INR-Bayes usually

reconstructs images closest to the ground truth. It forms a prior that contains color lumps similar to `MAML` as well as a face-like feature like `FedAvg`.

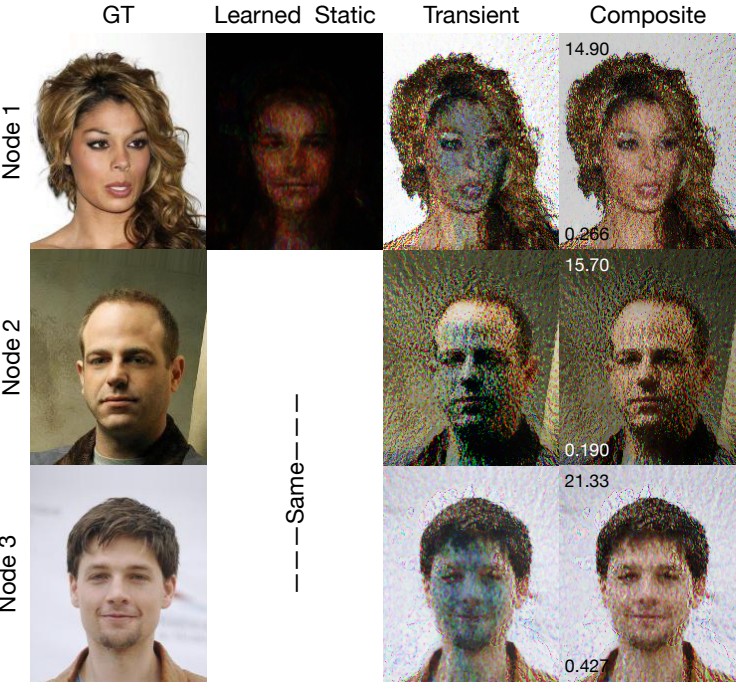

Figure 14: Learned static and transient parts of INRWild on human faces of CelebA dataset.

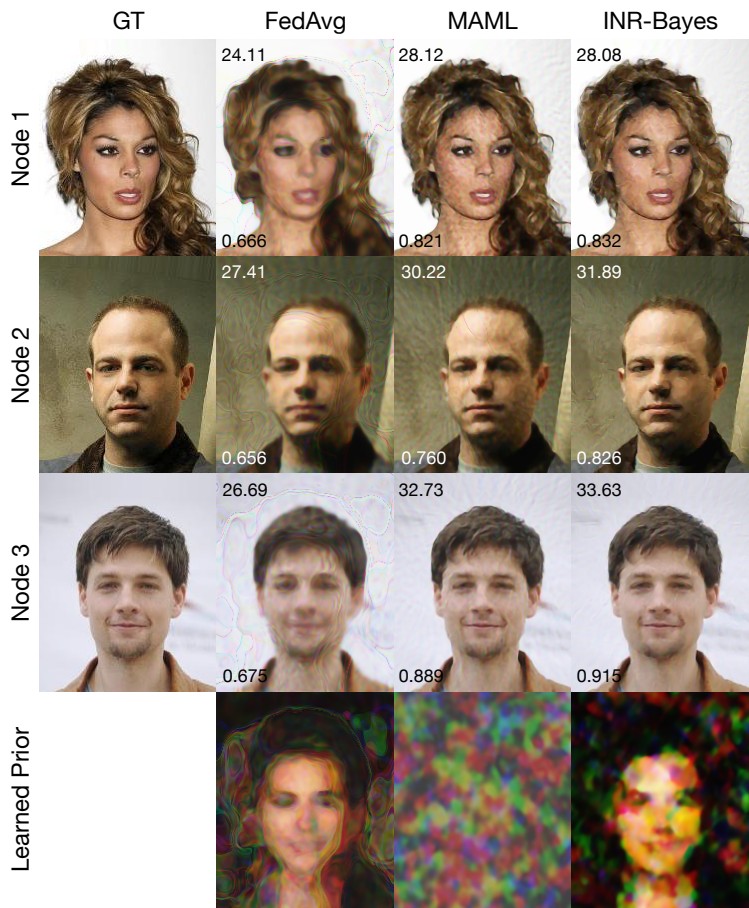

Figure 15: Visualization of the learned prior of joint reconstruction methods on the faces of the CelebA dataset.

### D.3 Inter-object Joint Reconstruction

In this section, we provide a visual comparison of inter-object joint reconstruction using the WalnutCT. As depicted in Figure 16, our approach exhibits the best PSNR and SSIM values, and delivers result that aligns better with the ground truth.

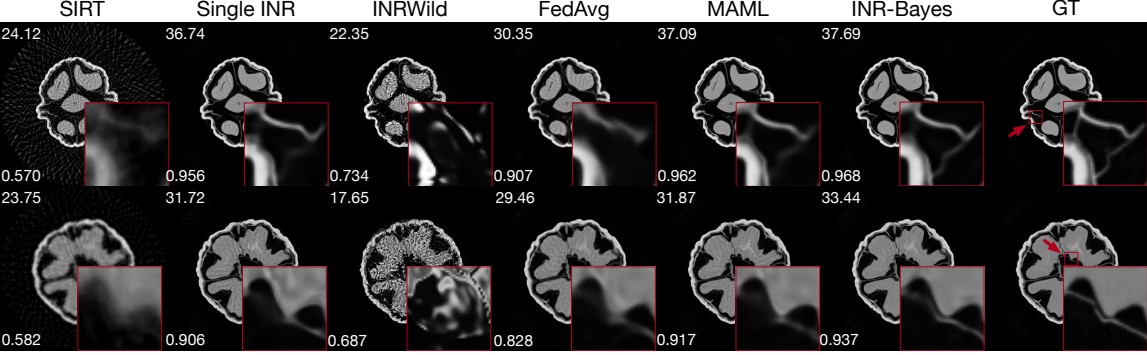

Figure 16: Visual comparison for inter-object walnut joint reconstruction. Enlarged areas are highlighted in red insets. PSNR values are on the top left, with SSIM values on the bottom left.

## D.4 Intra-object Joint Reconstruction

In addition to the experiments represented by Figure 10, we evaluated how the number of nodes influences the performance of various methods, in the intra-object joint reconstruction experiments. As shown in Figure 17, our method consistently delivers superior performance compared to other methods across a range of node counts. `MAML` shows strong results when the node count is between 5 and 25, but experiences a decline in performance, eventually matching that of `FedAvg` when the node count reaches 40. This drop indicates that `MAML` might struggle to capture the shared features when many nodes are participating in the joint reconstruction. Contrary to the inter-object configuration (as illustrated in Figure 10), in this experiment our method does not benefit from an increase in the number of joint nodes. This lack of improvement could be attributed to the possibility that the statistical regularities are sufficient when observing just a few scanning slices in the intra-object setting.

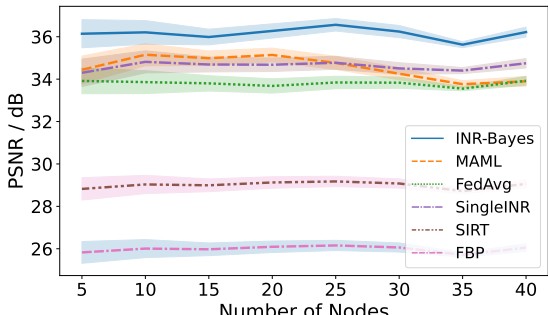

Figure 17: Performance comparison with varying node numbers in intra-object LungCT setup. Individual reconstruction methods are presented as references. Performance stability across all methods with increasing node numbers suggests that even a limited number of intra-object slices adequately capture statistical trends.

We also present visualizations of learned priors from various joint reconstruction methods applied in intra-object experiments on LungCT dataset. Figure 18 depicts the learned static and transient components of `INRWild`. Notably, in scenarios where images markedly vary from one another, extracting static components still seems feasible but not necessarily beneficial to the reconstruction process, since the static component does not constitute a significant portion of the overall representation.

Conversely, when observing other joint reconstruction methods in Figure 19, it is evident that `INR-Bayes`, `MAML` and `FedAvg` ascertain a reasonable latent or meta representation. Despite variations in images, the intrinsic consistency stemming from the same patient results in a discernible and coherent trend. This inherent trend is adeptly captured by these joint reconstruction methodologies.

## D.5 Joint Reconstruction across Temporal Phases in 4DCT

Figure 21 reveals that `INRWild` effectively distinguishes static and transient components in the 4DCT setting. Compared to other settings, `INRWild`'s performance on 4DCT is closer to other INR-based methods (see Table 1). However, it still lags behind `SingleINR`, underscoring the limited utility of static-transient tactics in enhancing individual reconstruction quality. Figure 22 demonstrates that other joint reconstruction methods also proficiently disentangle the inherent prior. Interestingly, in these contexts, `FedAvg` proves more beneficial than `MAML`, while our proposed method continues to surpass all methods in this scenario.

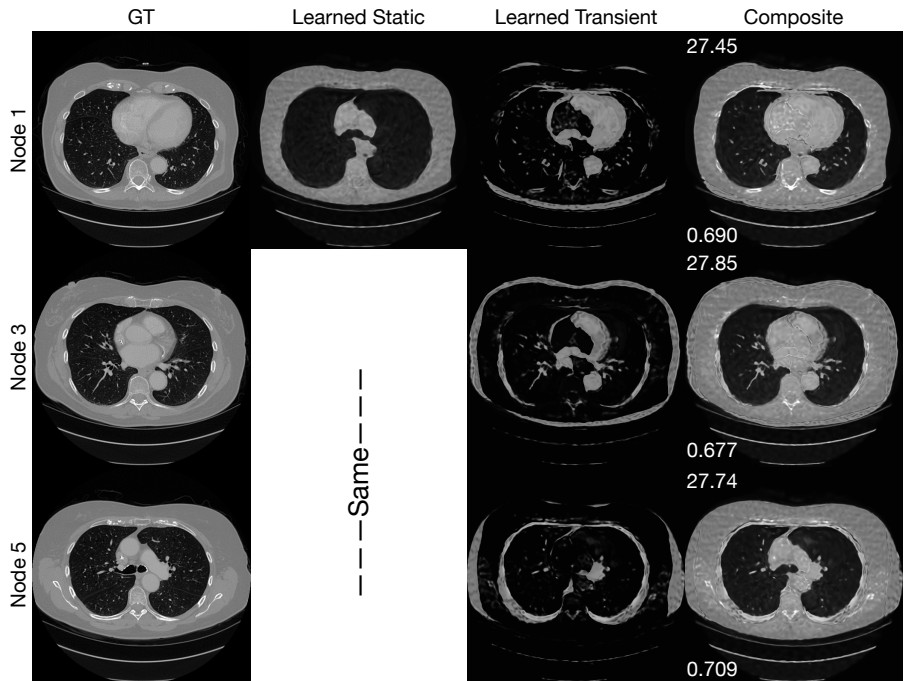

Figure 18: Learned static and transient parts of INRWild on LungCT dataset on the same patient.

### D.6 Joint Reconstructions with Varied Object and Slice Configurations

To understand how different joint reconstruction configurations impact reconstruction quality, we examine four distinct setups: inter-object (10 objects, each contributing one slice), intra-object (one object contributing ten slices), and two intermediate configurations (two objects with five slices each, and five objects with two slices each). We conducted ten sets of experiments, with each set involving the joint reconstruction of the same ten slices across these configurations.

Table 5 presents the average PSNR and SSIM values obtained from these ten experimental groups. The results indicate that while the PSNR and SSIM values are generally consistent across different configurations, the intermediate configurations slightly outperform the others in terms of average metrics.

Figure 23 illustrates the learned priors from one exemplary set of reconstructions. Visually, the distinctions among the different configurations are minimal. The intra-object configuration ($1\times10$) effectively captures the internal variations within a single object, aligning with our observation that a few intra-object slices are sufficient to model internal object changes. On the other hand, the $2\times5$ and $5\times2$ configurations, along with the inter-object ($10\times1$) setup, depict priors that appear more distinct from the target slice, reflecting the influence of multiple objects in the joint reconstruction process. Notably, the $2\times5$ and $5\times2$ configurations exhibit slightly higher PSNR and SSIM values, suggesting that an optimal balance between similarity and diversity among jointly reconstructed objects can enhance the reconstruction quality of a specific target.

| Config (*object × slices*) | $1 \times 10$ | $2 \times 5$ | $5 \times 2$ | $10 \times 1$ |
|---|---|---|---|---|
| PSNR | $35.56_{\pm1.03}$ | $35.76_{\pm1.08}$ | $35.66_{\pm1.07}$ | $35.59_{\pm1.02}$ |
| SSIM | $0.876_{\pm0.025}$ | $0.877_{\pm0.025}$ | $0.878_{\pm0.025}$ | $0.876_{\pm0.025}$ |

Table 5: Results from joint Reconstructions with varied object and slice configurations.

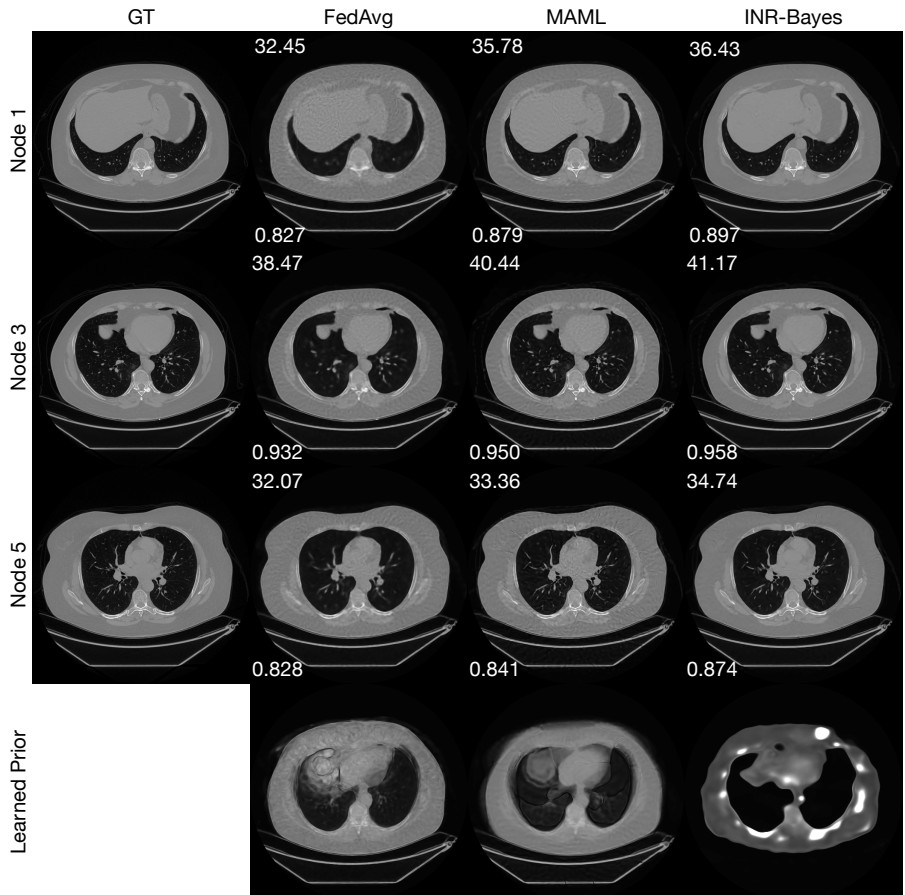

Figure 19: Visualization of the learned prior of joint reconstruction methods on the same patient of LungCT dataset.

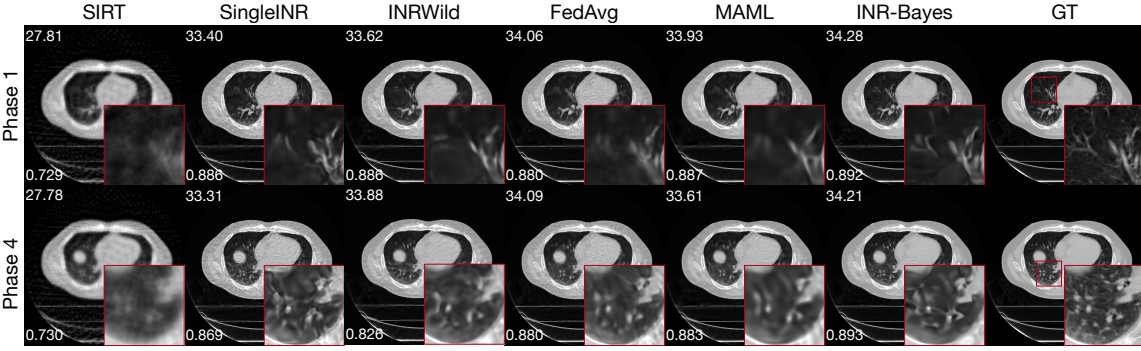

Figure 20: Visual comparison for joint reconstruction across 4DCT temporal phases. Enlarged areas are highlighted in red insets. PSNR values are on the top left, with SSIM values on the bottom left.

## D.7 Computation Cost Analysis

In Table 6, we present a comparative analysis of the computational costs associated with different reconstruction methods. The experiment setting is aligned with the inter-object configuration on WalnutCT dataset in Table 1. These assessments are performed under the same conditions with a 40GB A100 GPU, to ensure consistency in our evaluation. Each method undergoes 30,000 iterations and employs an identical Siren network architecture, characterized by a depth of 8, width of 128, and positional embedding dimension of 256.

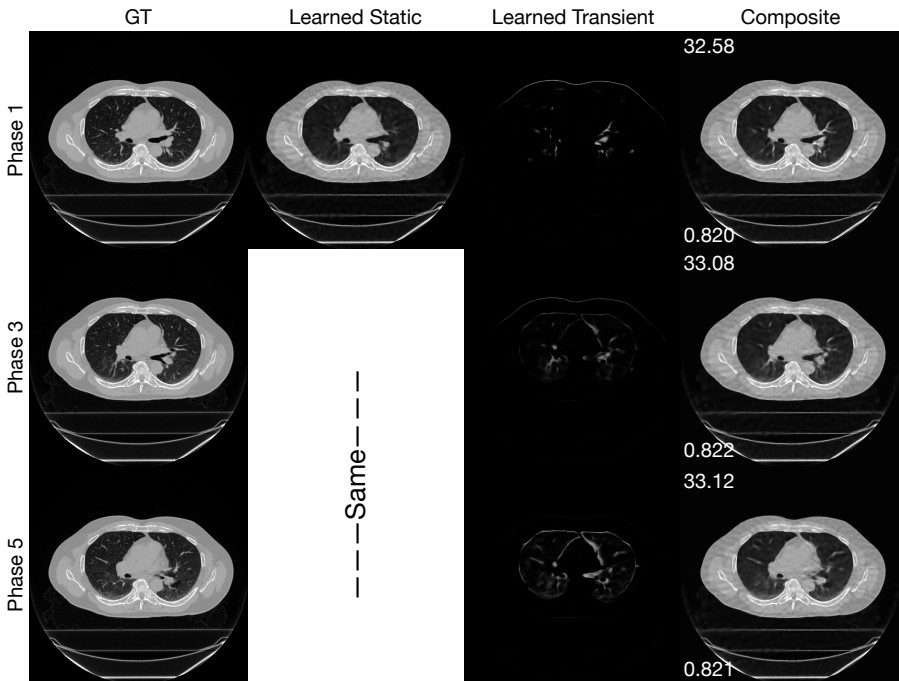

Figure 21: Learned static and transient parts of INRWild on 4DCT dataset.

|  | SingleINR | FedAvg | MAML | INR-Bayes |
|---|---|---|---|---|
| GPU Memory (MiB) | 3852 | 3972 | 3954 | 3990 |
| Time (mins) | 17.2 | 17.2 | 17.2 | 22.7 |

Table 6: Comparison of average computation cost per node on joint reconstruction of 10 nodes. The reconstructed image size is $501 \times 501$. For reference, FBP requires 206 MiB of GPU memory and completes in less than 0.1 seconds, whereas SIRT utilizes 154 MiB of GPU memory and takes 0.8 seconds.

As indicated in the Table 6, the GPU memory usage across all INR-based methods was relatively similar. `SingleINR`, `FedAvg` and `MAML` exhibit the same shortest computation time. Our method, `INR-Bayes`, shows a slightly increased computation time, approximately 5 minutes longer than the others. This less than 15% increase over `SingleINR`, `FedAvg` and `MAML` in time is attributed to the MC sampling procedure ($\sim 9\%$) and additional KL divergence term ($\sim 6\%$) in `INR-Bayes`. However, considering the enhanced reconstruction quality and robustness achieved, this additional time investment can be justified.

**Impact of Network Size on INRBayes.** To understand the computation efficiency and performance trade-off, we further investigate the reconstruction quality of our method with varying configurations of the network size. As shown in Table 7, smaller networks lead to reduced performance but also lower computational requirements, presenting a practical trade-off between efficiency and quality. It is worth noting that even with the smallest network, our method largely outperforms SIRT. While the reconstruction time using the smallest network is reduced to 10 minutes, which could be affordable in the most practical settings.

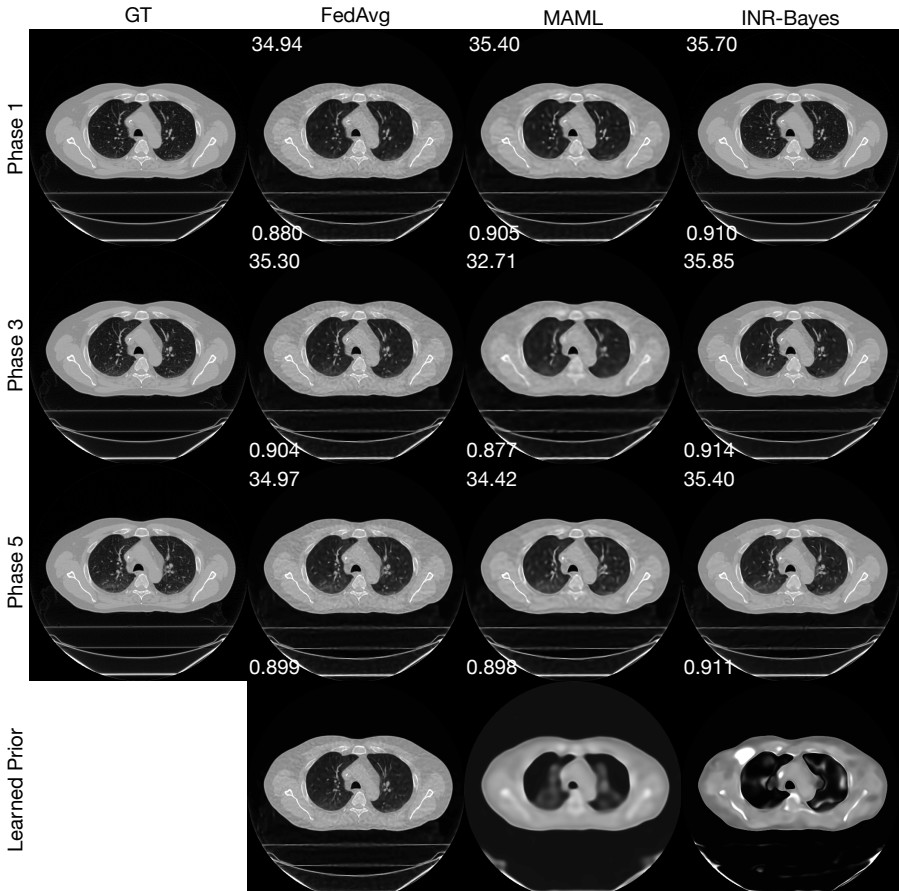

Figure 22: Visualization of learned prior of joint reconstruction methods on 4DCT dataset.

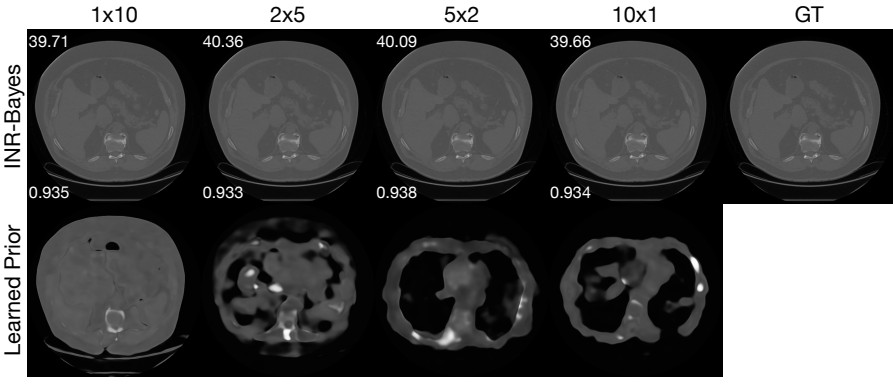

Figure 23: The reconstructions and learned priors with varied object and slice configurations.

| Depth | Embedding Size | PSNR | SSIM | Memory (MiB) | Time (mins) |
|-------|----------------|------|------|--------------|-------------|
| 8 | 256 | $35.45 \pm_{0.35}$ | $0.944 \pm_{0.003}$ | 3990 | 22.7 |
| 8 | 128 | $33.20 \pm_{0.49}$ | $0.918 \pm_{0.006}$ | 3900 | 22.5 |
| 6 | 256 | $33.75 \pm_{0.38}$ | $0.930 \pm_{0.005}$ | 3468 | 18.3 |
| 4 | 256 | $30.55 \pm_{0.37}$ | $0.906 \pm_{0.006}$ | 2980 | 10.6 |

Table 7: Impact of the network size on computation cost and performance of our method. For comparison, SIRT achieves a PSNR of $23.42 \pm_{0.21}$ and SSIM of $0.445 \pm_{0.004}$. The experiment setting is inter-object WalnutCT with 30,000 total iterations.

### D.8 Comparison with Nerp

The method `Nerp`, introduced by Shen et al. (2022), initially trains an INR network using high-fidelity data through regression. This pre-trained network is subsequently utilized to initialize the reconstruction of a new object with sparse measurements. A notable drawback of this method is its dependence on the new objects' representations being highly similar to that of the high-fidelity training object. When this similarity is absent, the initial training could hinder rather than help the reconstruction process, potentially yielding worse results than even a random initialization.

To demonstrate this, we carried out experiments on the 4DCT dataset with two different setups for `Nerp`. In the 'Match' configuration, `Nerp` is provided with the ground truth of one phase at a specific slice and tasked to reconstruct the remaining nine phases at that slice. In contrast, the 'Unmatch' configuration uses the ground truth from a random slice. Our `INR-Bayes` approach, on the other hand, performs a simultaneous reconstruction of all nine phases without any access to ground truth images.

As Table 8 illustrates, the performance of `Nerp` is conditional, excelling in PSNR when ground truth data is matched but faltering otherwise. While operating without access to additional information, our `INR-Bayes` achieves the best performance in SSIM. Given the practical challenges in obtaining matched ground-truth data for unscanned objects, our method exhibits greater utility and applicability.

| | SingleINR | Nerp match | Nerp unmatch | INR-Bayes |
|---|---|---|---|---|
| PSNR | 33.69 $\pm 0.06$ | **35.33** $\pm 0.10$ | 32.83 $\pm 0.07$ | 34.31 $\pm 0.07$ |
| SSIM | 0.883 $\pm 0.002$ | 0.889 $\pm 0.001$ | 0.849 $\pm 0.02$ | **0.901** $\pm 0.001$ |

Table 8: Comparison of Nerp performance in scenarios where the prior image either matches or does not match the target in the 4DCT dataset. The highest average PSNR/SSIM values that are statistically significant are highlighted in bold.

### D.9 Applying to Unseen Data using Different Learned Priors

To investigate the influence of varying priors on the reconstruction quality for new, unseen patients, we conducted an additional experiment with our `INR-Bayes`. We selected 10 sets of priors, each derived from a group of 10 different patients. These priors are then employed to guide the reconstruction of the same unseen patient. Figure 25 showcases the reconstructed images and their corresponding priors, represented by an INR that is parameterized with the mean of the prior distribution. The accompanying PSNR and SSIM values, indicated at the top left and bottom left of each image, demonstrate modest deviation across different priors. Notably, no model collapse occurs despite the obvious visual difference in the prior means. This observation suggests that our method is stable and can adaptively extract useful information from various priors when applied to unseen data.

It is important to clarify that the prior of our method, as depicted in Figure 25, uses the mean of the prior distribution to parameterize an INR. However, this representation is an incomplete portrayal of the prior distribution's full characteristics, as the latent variables include mean and variance estimations. The variance associated with our method's estimates can contribute to the adaptive and effective utilization of the prior distribution. This aspect of our model underscores its capability to leverage the entire prior distribution for stable performance.

### D.10 Reconstruction Uncertainty

We employ a Bayesian framework to identify common patterns among jointly reconstructed objects. While the primary focus of this study is not on uncertainty quantification (Gal & Ghahramani, 2016; Angelopoulos et al., 2022), we demonstrate how `INR-Bayes` can facilitate this aspect within CT reconstructions. During training, each node approximates the posterior distribution of its weights as a Gaussian distribution. This approach allows us to sample the network parameters multiple times after training to quantify uncertainty. To illustrate this, we sample the network parameters ten times to generate ten distinct reconstructions in an intra-object

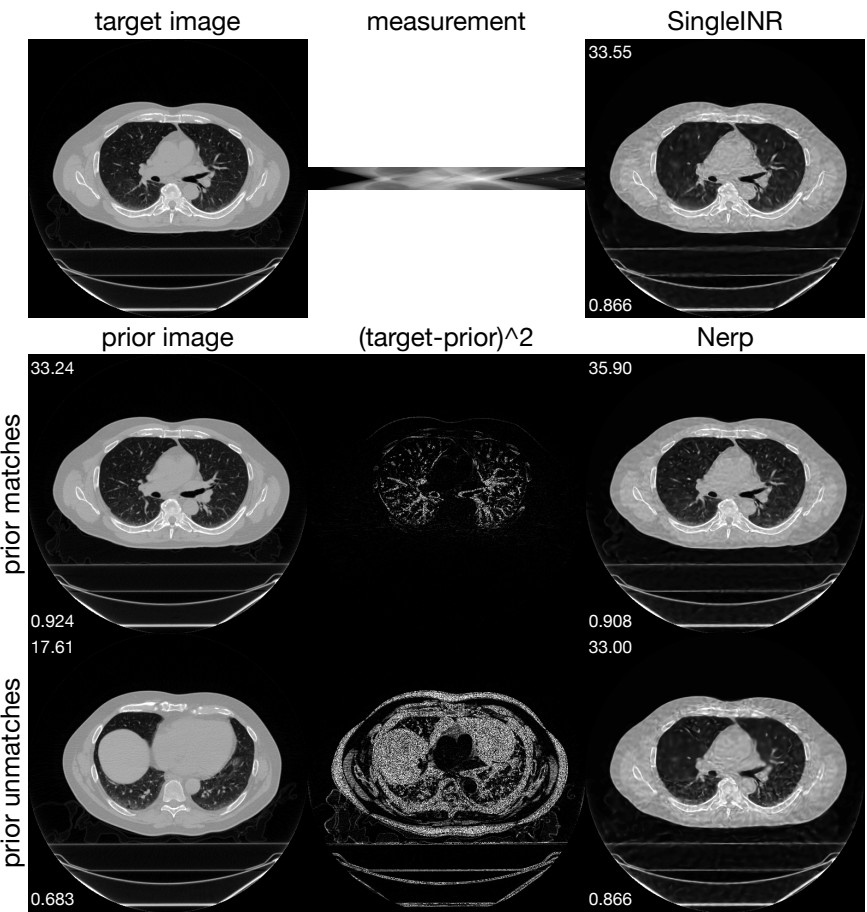

Figure 24: Results of Nerp with matched prior image and unmatched prior image. PSNR on the upper left corner and SSIM on the lower left corner are calculated with respect to the target image.

setup using the LungCT dataset. We then compute the mean and variance of these reconstructions to show the expected reconstruction and characterize the associated uncertainty in Figure 26.

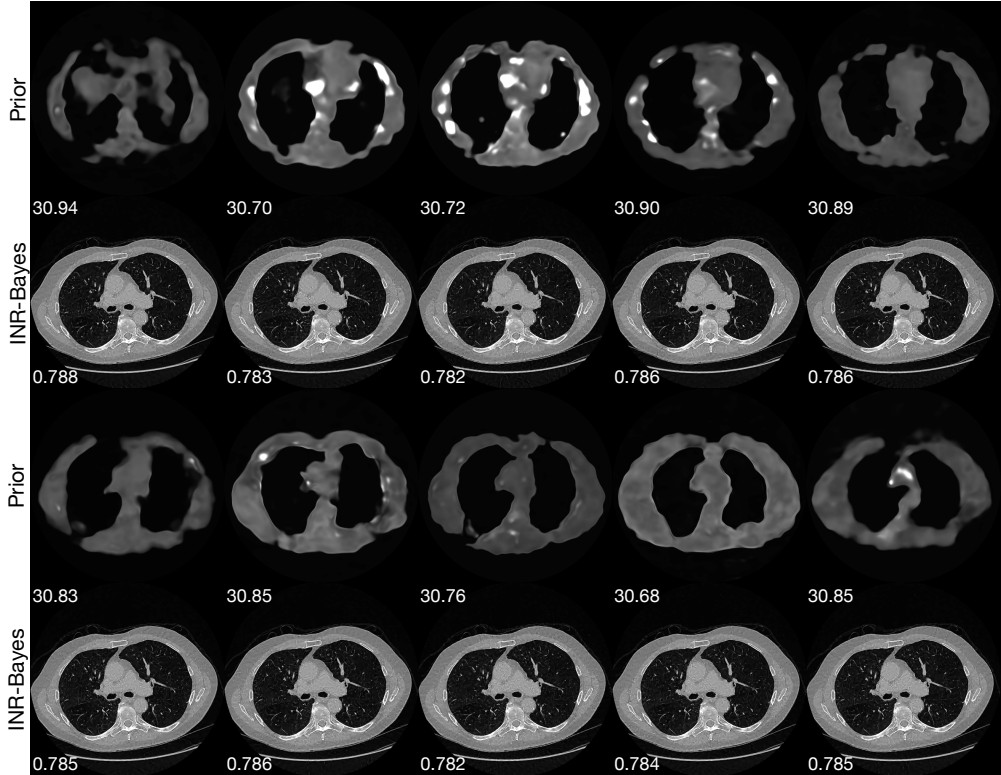

Figure 25: Reconstruction of the same unseen patient using different priors learned from various patient groups. The PSNR and SSIM values are presented on the top left and bottom left of each image, respectively, illustrating our method's robustness across different priors.

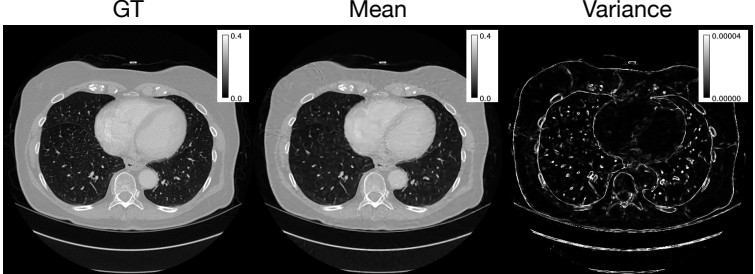

Figure 26: Mean and variance of 10 reconstructions by INR-Bayes.

