# OpenReview forum: "Implicit Neural Representations for Robust Joint Sparse-View CT Reconstruction"
_TMLR — Accepted by TMLR_

### Review · Reviewer_bPhg · 2024-05-29

**Summary Of Contributions:**

The authors propose to use implicit neural representations to improve reconstruction quality through joint reconstruction of multiple objects using implicit neural representations. They introduce a Bayesian framework that integrates latent variables to capture relationships between objects. The paper focuses on reconstructing sparse-view CT images and compares the method with other techniques, including Filtered Back Projection (FBP), iterative methods, some existing INR-based joint reconstruction techniques, a few-shot learning approach (MAML), and a federated learning approach (FedAvg). They demonstrate the results on several CT datasets, a synthetic dataset.

**Audience:**

Yes

**Broader Impact Concerns:**

No concerns.

**Claims And Evidence:**

No

**Requested Changes:**

- Comprehensive benchmarks should include state-of-the-art regularization-based approaches such as total generalized variation, wavelet regularization, and patch-based tensor representations (e.g. https://onlinelibrary.wiley.com/doi/full/10.1002/mrm.27694).
- Demonstration of broader impact on some real-world scenarios is necessary to validate the method's general applicability.

**Strengths And Weaknesses:**

**Strengths:**

- The method is novel and introduces an interesting approach to using implicit neural representations for joint reconstruction.
- Comprehensive ablation studies are conducted, thoroughly examining potential pitfalls such as overfitting, convergence issues, and computation time.
- The paper is well-motivated, clearly written, and has a very clear and logical structure, making it easy to follow and understand.
- Limitations of the method are clearly discussed.

**Weaknesses:**
- The benchmarks are limited and do not cover standard approaches, such as classical regularization methods, which are essential for a comprehensive evaluation (e.g. total generalized variation, wavelets regularization, patch-based tensor representations).
- The authors use RGB images "subjected to CT" as a "broader application" example. However, this example is artificial and does not convincingly demonstrate the broader applicability of the method. For a proper evaluation of the broader application, the authors should demonstrate its performance in real-world scenarios such as MRI or optoacoustics.
- Some claims such as for example: "This computational overhead yields **substantial** gains in reconstruction quality and robustness across diverse settings" are too strong according to the results tables and figures and should be toned down.

---

> ### Author Response · Authors · 2024-07-29
> **Response to Reviewer bPhg**
>
> Thank you for your thorough review and constructive comments! We appreciate the recognition of our paper's novelty and clarity. We address your feedback below:
>
> > Comparison with regularization-based methods
>
> We have conducted additional experiments to include several established regularization-based CT reconstruction methods [Niu et al, 2014; Garduno et al, 2011; Kazantsev et al, 2017]. The factor for regularization strength is determined by grid searching for the highest PSNR values on holdout data. The experiment results show that these methods generally yield lower PSNR and SSIM across CT datasets.  Upon reviewing the visualizations, we observe that reconstructions from these methods often appear over-smoothed, with a noticeable reduction in detail compared to our proposed method.
>
> We have revised our manuscript to include new results and findings (in Table 1, Figure 3, Figure 6, Section 5.1 subsection “Reconstruction Performance”).
>
>
> > Broader application
>
> We acknowledge the critique regarding the use of RGB images "subjected to CT" as a demonstration of broader applicability. In response, we have adjusted our manuscript to remove the subsection on broader application. The results from the CelebA dataset are now presented solely within the Section 5.1 "Reconstruction Performance" subsection. We consider expanding this method to other modalities as future work. In this work, our claims and evidence primarily focus on CT reconstruction on the demonstrated datasets.
>
> > Revision of claims
>
> We have revised the sentence highlighted by the reviewer to tone down the language. Additionally, we have reviewed our results tables and figures, carefully moderating the language used to describe our findings to ensure it accurately reflects the demonstrated outcomes.

---

> > ### Comment · Reviewer_bPhg · 2024-08-19
> >
> > The authors addressed all my concerns.

---

### Review · Reviewer_2CxD · 2024-06-13

**Summary Of Contributions:**

The submission proposes a novel Bayesian approach for the joint reconstruction of several objects from sparse-view CT measurements. The proposed method consists of training implicit neural representations (INRs) concurrently---one per object being reconstructed---by coupling them with a latent prior that should encode common information across objects. Experiments and ablation tests are carried out to compare the proposed method with existing INR-based alternatives on 4 different datasets coming from different domains.

**Audience:**

Yes

**Claims And Evidence:**

Yes

**Requested Changes:**

**Disclaimer on use of language models**

I use GPTZero to check all abstracts of submissions I review. For this submission, the abstract was reported as 100% AI-generated. I would urge authors to review TMLR's guidelines on the disclosure of the use of language models, available at https://jmlr.org/tmlr/faq.html .

---

**Abstract**

Use of jargon makes the text unintuitive to people not familiar with the topic. In particular:

* "we introduce a novel ... the inter-object relationships" is hand-wavy before reading the paper.
* "meta-initialization", "dynamic reference", "individual reconstruction fidelity", "resistance to overfitting", and "generalizability" are all used without definition. These are very broad terms with different meaning and possible interpretations.

I would suggest to smooth the claim:
> This underscores a notable advancement in CT reconstruction methods.

---

**Introduction**

> posing a complex inverse problem.

"complex" may be mistook here for "complex number" instead of "complicated".

> However, this sparsity in measurements complicates the reconstruction process, making it an ill-posed inverse problem.

Ill-posedness of the problem does not stem only from the use of Sparse-view CT but also other factors such as number of detectors, geometry of the scanner, and the presence of noise.

> ... to numerous inverse problems.

It may be helpful to include a couple examples.

> Supervised learning ... Alternatively, Song et al ...

It may be useful to rephrase this paragraph to stress that supervised learning and diffusion models are not the only two perspectives on CT reconstruction. Also, several works have specialized the fundamental ideas of Song et al. to CT applications, and they may be worth mentioning here for a fair representation of the research landscape.

- Typo in "there are also some studies leverage".

> On a different tangent ...

This makes it sound as if what comes next is only indirectly related to the problem of CT reconstruction, but the paragraph introduces INR-based methods, which are at the core of the contributions of the submission.

> Thanks to the continuous representation nature of INRs ...

I am not sure I follow how continuity leads to better reconstruction results, there may be other factors in support of INRs?

> Given INRs' ... and the known advantages of leveraging auxiliary information ...

It is unclear what "auxiliary information" and "advantages" refer to here.

> ... among different objects borne in the INR networks' weights ... cannot fully capitalize on the available statistical regularities ...

It is unclear to me what this sentence means. What does it mean for an object to be borne in the weights of a network? How can one measure if a method is "fully capitalizing" compared to "partially capitalizing"?

---

**Related work**

> ... with a comprehensive understanding on INR and NeRF available in the survey (Tewari et al., 2022).

INRs are at the core of the contributions of the submission, but they are not formally introduced. A more careful introduction of these ideas is necessary to make the submission accessible to readers coming from the broader CT reconstruction field who may not necessarily know about INRs.

Furthermore, the cited survey "[does] not focus on neural rendering methods that reason mostly in 2D screen space". It is my understanding that the objects studied in this submission are 2D images. Finally, "the authors [of the survey] do not claim completeness of the report and highly recommend the reader to study the cited works for in-depth details", so I would suggest introducing the core ideas directly in the main text of the submission and including more foundational introductory references for interested readers who might want to learn the details.

> Coordinates-based ... have transitioned from traditional discrete representations ... detailing issues.

It may be helpful to expand on this sentence. In its current form, it is unclear what "discrete representations" are (i.e., pixels on a grid). "have transitioned" is a very broad claim, and traditional MLPs are still prominent in current research. Finally, without further explanation, it is unclear what "addressing high-frequency function detailing issues" means.

Onward, this paragraph describes concepts and ideas related to scene understanding (e.g., "viewing angles", "transmittance effects", "ray-tracing", "video-representation", ...). It is unclear to me how these topics are relevant to set the stage for the presentation of the main contributions of the submission, and, to someone without specific scene understanding knowledge, they are confusing.

---

**Problem Statement and Preliminaries**

> ... where $\epsilon$ accounts for the associated measurement noise.

It may be helpful to specify that $\epsilon$ is a stochastic quantity. Could the authors expand on what kind of noise they consider? In this form, it seems that $\epsilon$ is independent additive noise (maybe Gaussian?), but later in Appendix B.1, Eq. (22) introduces Poisson noise, which is not additive.

> [the INR designed for CT reconstruction] maps ... to its intensity in a continuous three-dimensional space.

It is unclear whether the "continuous three-dimensional space" is the input or output domain of the INR. Intensity is a scalar value.

> ... and measurement matrices $A_{1:J}$.

The joint reconstruction problem is introduced with different measurement matrices. Could the authors expand on whether in their experiments all objects being reconstructed have the same measurement matrix $A$, or how it differs across objects.

> ... can provide a more principled approach.

Could the authors expand on in what sense is including a dynamic prior shared across nodes more principled?

> ... and corresponding transient feature $b_j$.

Could the authors expand on whether these features are learned, and if so how?

> Here, $\mathcal{G}_{\phi}^{r}(C)$ ... and $\mathcal{G}_{\phi}^{\r}(C)$ ...

These symbols are neither defined in the main text, nor in Appendix B.2 .

> ... characterized ... characteristics.

Note the repetition. What characteristics of $\phi$ are being referred to here?

* Eq. (4):
    - It may be helpful to use the same notation $w^{(k)}$ and $\theta^{(t)}$ instead of $\theta^t$.
    - Is there a missing transpose in Eq. (4) between the gradient and the Jacobian matrix?

---

**A Novel Bayesian Framework for Joint Reconstruction**

> ... all representation substantially overlap?

Could the author expand on what it means for representations to overlap.

* Role of transient features in proposed method

It was unclear to me whether transient features are used in the proposed method, and, if so, how they are learned within the Bayesian framework.

> While this posterior enables various forms of deductive reasoning.

I am not sure I follow the meaning of this sentence in the context of the paragraph.

> ... that aims at maximizing the marginal likelihood $p(y_{1:J} \mid \omega, \sigma)$.

Why is this term called marginal likelihood if it is conditional on the latent variables?

> ... parameterizing the variance $\sigma_j$ ...

Typo in parametrizing. Also, should this be $\rho_j$ instead of $\sigma_j$?

* Eq. (5) and (7)

I would suggest to maintain consistent notation of the inputs of the ELBO function across equations.

* Questions on proposed method

It may be useful to remark that the latent space is as big as the nodes. It is common for latent spaces to have a dimension smaller than the object of interest. Here, the latent space keeps track of the mean and variance of the nodes, so its dimension is fixed.

Could the authors expand on whether the proposed method can be extended to latent spaces smaller than the nodes? For example, what would happen if the latent space were as large as the images being reconstructed? Would this retrieve a similar approach to INRWild?

Maybe, it could be interesting to include some example images generated from the learned prior, since the prior is an INR itself.

Could the authors expand on the difference between the proposed method and the federated learning approach FedAvg? The latent mean $\omega$ can be interpreted as a meta network, so the main advantage is in considering the variance as well?

Similarly, all nodes are assumed to be the same size, but this assumption is not stated clearly in the text. I would suggest including this remark. Could the proposed method entail nodes of different size? What would the main hurdles to achieve that be?

Finally, could the authors expand on the choice of using the posterior expectations instead of sampling? An advantage of Bayesian methods is to include uncertainty quantification out-of-the box, and it would be interesting to compare with diffusion-based methods.

---

**Experiments**

* Missing reference for SIRT:

Gilbert, Peter. "Iterative methods for the three-dimensional reconstruction of an object from projections." Journal of theoretical biology 36.1 (1972): 105-117.

* Typo in "for the rest experiments"

* CelebA experiments:

It is unusual to see the CelebA dataset with a CT forward model. Have the authors considered showcasing their proposed method for a different, more common task on the CelebA dataset, such as image denoising or inpainting? This may better place the proposed method in comparison with existing denoising techniques that also do not rely on the availability of large datasets.

* Clearly state if results are on noiseless reconstruction

CT reconstruction is presented as a noisy inverse problem. However, Table 1 shows results for noiseless measurements. This should be stated clearly before presenting results.

Could the authors expand on their motivation for comparing on all settings for noiseless measurements and only on the inter-object setup for noisy measurements?

> ... overfitting when applied to limited data.

Could the authors expand on this claim? Iterative reconstruction methods do not require a trained predictor, so what does "limited data" refer to here? What kind of overfitting is being referred here since measurements are noiseless at this point in the submission. In a noiseless setting, this may be an issue of stability and convergence of the optimization process rather than overfitting?

* It is never specified which method the curves in Figure 7 are generated with. Similarly to my confusion above, Figure 7 is used as evidence of overfitting in the noisy setting, but similar claims are also made with Figure 4 in the noiseless setting.

> ... we select 5 consecutive slices from new objects, choosing slices from a similar location the prior has been trained.

It is not specified how many times this process was repeated to obtain confidence intervals. What does "similar location" mean? Is there a particular tolerance that was used to include slices from different objects?

* Broader applicability

I assume this paragraph is talking about the CelebA dataset, although it is never defined what "natural RGB images" are considered, beyond mentioning "faces". If this paragraph is about the CelebA experiment, I would suggest moving it before Table 1 is presented because it includes details about how those experiments are carried out.

---

**Conclusions**

> ... plagued by the challenges of sparse measurements.

I would consider rewording this sentence.

**Strengths And Weaknesses:**

Strengths:
- Efficient and fast reconstruction of several objects from CT measurements is a relevant problem.
- The proposed method is novel.
- Results show improvements compared to existing methods.

Weaknesses:
- Extensive use of jargon makes presentation less accessible to first-time readers
- Notation and presentation of main contributions is somewhat hand-wavy
- Certain claims could be made clearer or reworded.

I will expand on my comments and I am looking forward to discussing with the authors to clarify my confusion.

---

> ### Author Response · Authors · 2024-07-29
> **Response to Reviewer 2CxD - Part 1**
>
> Thank you for your comprehensive review and constructive feedback! We appreciate your recognition of the novelty and relevance of our work. In response to your suggestions, we have carefully revised our manuscript to enhance clarity and accessibility, aiming to make our research more approachable to a broader audience. Below, we address your feedback in detail:
>
> > Disclaimer on use of language models
>
> We acknowledge the use of language models to assist in refining the manuscript, specifically for tasks like grammar corrections and rephrasing sentences. To align with transparency guidelines, we have now included a disclaimer in the footnote on the first page stating, "We have used ChatGPT provided by OpenAI to assist in writing. The language model was employed at the sentence level for tasks such as fixing grammar and rewording sentences. We assure that all ideas, claims, and results presented in this work are human-sourced."
>
> > "complex" may be mistook here for "complex number" instead of "complicated".
>
> We have changed the word from “complex” to “complicated” (Section 1 introduction paragraph 1).
>
> > However, this sparsity in measurements complicates the reconstruction process, making it an ill-posed inverse problem.
> Ill-posedness of the problem does not stem only from the use of Sparse-view CT but also other factors such as number of detectors, geometry of the scanner, and the presence of noise.
>
> We agree that many factors can lead to ill-posedness. Therefore, we have revised the sentence to “However, this sparsity in measurements, along with other factors such as the presence of noise and the size of detectors, complicates the reconstruction process, making it an ill-posed inverse problem.” (Section 1 introduction paragraph 1)
>
> > ... to numerous inverse problems.
> It may be helpful to include a couple examples.
>
> We have added a couple of examples. This part now reads: "to numerous inverse problems, such as Magnetic Resonance Imaging and Ultrasound Imaging." (Section 1 introduction paragraph 1)
>
> > Supervised learning ... Alternatively, Song et al ...
> It may be useful to rephrase this paragraph to stress that supervised learning and diffusion models are not the only two perspectives on CT reconstruction. Also, several works have specialized the fundamental ideas of Song et al. to CT applications, and they may be worth mentioning here for a fair representation of the research landscape.
>
> We agree that supervised learning and diffusion models are not the only perspectives on CT reconstruction. We have added the sentence: “It is important to note that supervised learning and diffusion-based approaches represent only a subset of the methods available for CT reconstruction.” (Section 1 introduction paragraph 2)
> Additionally, we have expanded our discussion to include specific adaptations of diffusion-based methods to CT applications. We have cited several studies [Chung et al., 2022; Liu et al., 2023; Song et al., 2023; Xu et al., 2024] that build upon the foundational ideas introduced by Song et al. [Song et al., 2024]. If the reviewer has specific references to highlight, we would be glad to review and cite them. The expanded sentence reads “Several diffusion-based approaches [Chung et al., 2022; Liu et al., 2023; Song et al., 2023; Xu et al., 2024] have been specifically tailored for CT applications, further refining the foundational ideas presented by Song et al. [Song et al., 2024].” (Section 1 introduction paragraph 2)
>
> > Typo in "there are also some studies leverage"
>
> We have added “that” into the sentence (Section 1 introduction paragraph 2)
>
> > On a different tangent ...
> This makes it sound as if what comes next is only indirectly related to the problem of CT reconstruction, but the paragraph introduces INR-based methods, which are at the core of the contributions of the submission.
>
> We have modified the sentence from “On a different tangent ...” to “Building on the foundations of CT reconstruction”. (Section 1 introduction paragraph 3)
>
> > Thanks to the continuous representation nature of INRs ...
> I am not sure I follow how continuity leads to better reconstruction results, there may be other factors in support of INRs?
>
> We acknowledge your point about the factors contributing to the performance of INRs. To address it, we have changed the sentence to “Thanks to the advantages of INRs, such as their continuous representation nature in contrast to conventional discrete representations [Lee et al, 2021; Xu et al, 2022; Grattarola & Vandergheynst, 2022]”. We have also added citations that discuss continuous representations. (Section 1 introduction paragraph 3)

---

> ### Author Response · Authors · 2024-07-29
> **Response to Reviewer 2CxD - Part 2**
>
> > ... among different objects borne in the INR networks' weights ... cannot fully capitalize on the available statistical regularities ...
> It is unclear to me what this sentence means. What does it mean for an object to be borne in the weights of a network? How can one measure if a method is "fully capitalizing" compared to "partially capitalizing"?
>
> In INR-based methods, the objects are essentially encoded within the learned network weights, which is what we referred to by stating they are "borne in the weights of a network."  For methods such as MAML and FedAvg, while they utilize these regularities for joint reconstruction, their primary focus often shifts towards enhancing convergence speed rather than maximizing reconstruction accuracy.
>
> We recognize that our initial wording might have caused confusion. To clarify, we have revised our manuscript to state: "These approaches have utilized the inherent statistical regularities across different objects, represented by the network weights in INRs. However, they often focus on addressing issues like convergence rate [Tancik et al., 2021; Kundu et al., 2022] rather than optimizing the reconstruction quality itself. As demonstrated in our experimental evaluations (section 5), these methods achieve lower PSNR and SSIM compared to our proposed method in joint reconstruction scenarios." (Section 1 introduction paragraph 3)
>
> > ... with a comprehensive understanding on INR and NeRF available in the survey (Tewari et al., 2022).
> INRs are at the core of the contributions of the submission, but they are not formally introduced. A more careful introduction of these ideas is necessary to make the submission accessible to readers coming from the broader CT reconstruction field who may not necessarily know about INRs.
>
>
> We recognize the importance of making our submission accessible to a broader audience, particularly those in the CT reconstruction field who may not be familiar with the specifics of INRs.
>
> To address this, we point it out here explicitly that the core INR for CT reconstruction idea and working principles are introduced in the following section "The key working principles of INR for CT reconstruction is explained in Section 3." Furthermore, to provide a non-technical overview, we have modified the paragraph for INR "Coordinate-based Multi-Layer Perceptrons (MLPs) intake spatial coordinates and output values, such as RGB or density, processed through MLP. Unlike traditional discrete representations that directly map inputs to outputs on pixel grids, these coordinate-based outputs are generated by the MLP. Therefore, such coordinate-based representations are often regarded as INRs. Originally, INRs faced challenges in capturing high-frequency details. However, the introduction of coordinate encoding, such as Fourier feature transformations [Jacot et al., 2018; Tancik et al., 2020], has significantly enhanced their ability to represent finer details." (Section 2 Related Work paragraph 2)
>
> > Coordinates-based ... have transitioned from traditional discrete representations ... detailing issues.
> It may be helpful to expand on this sentence. In its current form, it is unclear what "discrete representations" are (i.e., pixels on a grid). "have transitioned" is a very broad claim, and traditional MLPs are still prominent in current research. Finally, without further explanation, it is unclear what "addressing high-frequency function detailing issues" means.
>
> We have expanded this sentence to a paragraph as mentioned in the previous response.  (Section 2 Related Work paragraph 2)

---

> ### Author Response · Authors · 2024-07-29
> **Response to Reviewer 2CxD - Part 3**
>
> > Onward, this paragraph describes concepts and ideas related to scene understanding (e.g., "viewing angles", "transmittance effects", "ray-tracing", "video-representation", ...). It is unclear to me how these topics are relevant to set the stage for the presentation of the main contributions of the submission, and, to someone without specific scene understanding knowledge, they are confusing.
>
> We acknowledge that these terms are primarily associated with optical scene rendering techniques like NeRF and may not directly relate to the traditional understanding of CT imaging.
>
> NeRF reconstructs the scene with INR from multiple photos taken from different angles. While fundamentally different in application, it shares similarities with CT reconstruction in how it processes data in terms of reconstruction. However, unlike NeRF, which handles light transmittance and occlusion to render RGB values, CT reconstruction leverages penetrating X-rays to compute attenuation coefficients, revealing the internal structure of objects.
>
> Recognizing the need for clearer exposition, we have revised the relevant section to better explain these concepts. We explicitly state why NeRF methodologies are discussed (Section 2 Related Work Paragraph 3, also copied below), emphasizing that some ideas from NeRF have been adapted and compared as baselines in our CT reconstruction framework.
>
> "NeRF is a state-of-the-art INR approach that reconstructs scenes from photos taken at multiple angles [Mildenhall et al., 2021; Barron et al., 2021; 2022], similar to CT where projections are acquired at different angles to compute reconstruction. While NeRF processes optical images to render RGB values by considering light transmittance and occlusion, CT reconstruction calculates attenuation coefficients to reveal internal compositions using penetrating X-rays. We therefore briefly introduce the NeRF methods that we adapt as comparison baselines. "
>
> > ... where $\epsilon$ accounts for the associated measurement noise.
> It may be helpful to specify that $\epsilon$ is a stochastic quantity. Could the authors expand on what kind of noise they consider? In this form, it seems that $\epsilon$ is independent additive noise (maybe Gaussian?), but later in Appendix B.1, Eq. (22) introduces Poisson noise, which is not additive.
>
> We acknowledge the need for clearer definitions regarding noise in our model. To address this, we have refined our description as follows: “The CT acquisition process, in an ideal noiseless scenario, can be mathematically described by the equation: $y = Ax$, where $x \in \mathbb{R}^m$ represents the unknown object of interest and $y \in \mathbb{R}^n$ symbolizes the ideal noiseless measurements. In practice, the actual obtained measurements $\widetilde{y}$ are often noisy, where the difference between actual and ideal measurements $\epsilon=\widetilde{y}-y$ is the measurement noise. We note that $\epsilon$ is a stochastic term representing general noise, including non-additive types like Poisson noise, which is the primary focus of this study.” (Section 3 paragraph 1)
>
> > [the INR designed for CT reconstruction] maps ... to its intensity in a continuous three-dimensional space.
> It is unclear whether the "continuous three-dimensional space" is the input or output domain of the INR. Intensity is a scalar value.
>
> We have modified the sentence to “The INR designed for CT reconstruction is a function $f_{w}: \mathbb{R}^3 \rightarrow \mathbb{R}^1$ parameterized by $w$. It maps the three-dimensional spatial coordinates of the object to its intensity.” (Section 3 paragraph 2)
>
> > ... and measurement matrices $A_{1:J}$.
> The joint reconstruction problem is introduced with different measurement matrices. Could the authors expand on whether in their experiments all objects being reconstructed have the same measurement matrix $A$, or how it differs across objects.
>
> In our manuscript, the formulation is presented in a general context to accommodate scenarios where different objects may have distinct measurement matrices $A_j$. However, for the experiments conducted in this study, we have focused on a scenario where all objects share the same measurement matrix $A$.
>
> To avoid confusion, we have clarified this in the manuscript by adding the following sentence:
> "While this formulation allows for different measurement matrices for each object, we primarily consider scenarios where all objects share the same measurement matrix $A$ in our experiments." (Section 3 paragraph 4)
>
> > ... can provide a more principled approach.
> Could the authors expand on in what sense is including a dynamic prior shared across nodes more principled?
>
> To avoid confusion, we have removed the sentence of “ can provide a more principled approach.”  (Section 3 paragraph 4)

---

> ### Author Response · Authors · 2024-07-29
> **Response to Reviewer 2CxD - Part 4**
>
> > ... and corresponding transient feature $b_j$.
> Could the authors expand on whether these features are learned, and if so how?
>
> These features $b_j$ are learned components within INRWild. As outlined in Formula 3 of our manuscript, $b_j$ is one of the optimization targets. This is demonstrated in Figure 12 of our manuscript, which corresponds to Figure 3 in the NeRFWild paper [Martin-Brualla et al., 2021]. Here, $b_j$ is fed along with outputs from the static network $\mathcal{G}_{\phi}$ into the transient MLP, and it is optimized jointly with the parameters of both the transient and individual MLPs.
>
> To improve clarity, we have revised the corresponding section in the appendix to elaborate on the learning process of $b_j$: "Each transient part is linked to a unique one-hot vector, which is embedded into a 16-dimensional transient feature $b_j$. This feature, alongside intermediate features $\mathcal{G}_{\phi}^{\backslash r}(C)$ from the static network, serves as input for transient modeling. The transient feature $b_j$ is optimized during joint training." (Appendix B.2)
>
> > Here, $\mathcal{G}_{\phi}^{r}(C)$ ... and $\mathcal{G}_{\phi}^{\r}(C)$ ...
> These symbols are neither defined in the main text, nor in Appendix B.2 .
>
> While we previously mentioned that "$\mathcal{G}_{\phi}$ represents the neural representation for the static component, and $\mathcal{H}{w}$ signifies the transient component," it appears this was insufficiently clear.
>
> To resolve any confusion, we have explicitly defined these components in the revised manuscript as follows:
>
> "$\phi$ represents the weights of the MLP $\mathcal{G}$.",
>
> "$w$ represents the weights of the MLP $\mathcal{H}$.",
>
>  "The output of the static network $\mathcal{G}_{\phi}$
>
> is split into two components: $\mathcal{G}_{\phi}^{r} (C)$ ,
>
> which represents the static intensity, and $\mathcal{G}_{\phi}^{\backslash r} (C)$, which serves as intermediate features for the transient networks. The symbols $r$ and $\backslash r$ denote the division of outputs into these two components."  (Section 3.1 paragraph 3)
>
> > ... characterized ... characteristics.
> Note the repetition. What characteristics of $\phi$ are being referred to here?
>
> To clarify, $\phi$ represents the parameters that define the static part of our model, which serves as a foundation for the dynamic components characterized by $w_{1:J}$.
>
> To remove redundancy and enhance clarity, we have revised the sentence to:
> "The individual representations, characterized by $w_{1:J}$, are refined to complement the foundational static framework defined by $\phi$." (Section 3.1 paragraph 3)
>
> > Eq. (4): It may be helpful to use the same notation $w^{(k)}$ and $\theta^{(t)}$ instead of $\theta^t$.
> Is there a missing transpose in Eq. (4) between the gradient and the Jacobian matrix?
>
> We have unified the notation and added the transpose.  (Section 3.1 paragraph 4)
>
> > ... all representation substantially overlap?
> Could the author expand on what it means for representations to overlap.
>
> NeRFWild addresses scene reconstruction by disentangling static and transient elements of scenes captured in the wild, such as popular tourist sites photographed from various angles and under different lighting conditions. The "overlapping" of representations refers to the core structural features of the scene—typically the primary objects like landmarks—that remain constant across different images despite variations in perspective, lighting, and occlusions by transient objects like pedestrians.
>
> To enhance clarity on this aspect, we have revised the corresponding section to state: "NeRFWild is proposed to address scene reconstruction challenges, such as reconstructing popular sightseeing sites from in-the-wild photos. These scenarios often involve a primary target object, like a landmark, amidst various transient elements such as pedestrians and changing light conditions. A method that uses a composition of static and transient components operates under the assumption that the core representations of the scene—those of the main object—substantially overlap, despite different viewpoints or transient changes. This assumption typically holds true in 3D scene reconstruction, where observations are collected from multiple perspectives of the same object." (Section 4 paragraph 2)
>
> > Role of transient features in proposed method
> It was unclear to me whether transient features are used in the proposed method, and, if so, how they are learned within the Bayesian framework.
>
> We would like to clarify that transient features pertain exclusively to INRWild, which we reference as a comparative baseline in our study. Unlike INRWild, our proposed method does not utilize a composition of static and transient components, and thus, does not involve learning transient features.

---

> ### Author Response · Authors · 2024-07-29
> **Response to Reviewer 2CxD - Part 5**
>
> > While this posterior enables various forms of deductive reasoning.
> I am not sure I follow the meaning of this sentence in the context of the paragraph.
>
> The term "posterior" here refers to the distribution over potential reconstructions conditioned on observed data, which is a central target in Bayesian inference approaches. By accurately estimating this posterior, one can derive various insights, such as uncertainty quantification and the integration of prior knowledge to enhance reconstruction quality.
>
> To address the concern and minimize confusion, we have expanded the statement in our manuscript: “While this posterior enables various forms of deductive reasoning such as reconstruction uncertainty quantification” (Section 4 paragraph 5)
>
>
> > ... that aims at maximizing the marginal likelihood $p(y_{1:J} \mid \omega, \sigma)$.
> Why is this term called marginal likelihood if it is conditional on the latent variables?
>
> In statistical terms, the "marginal likelihood" refers to the likelihood obtained by integrating over the parameters. In our context, while $w_{1:J}$ are parameters whose distribution we infer, we actually focus on the likelihood of the observed data after integrating out these parameters, hence the term "marginal." We have revised the sentence to clarify it,  “that aims at maximizing the marginal likelihood $p(y_{1:J} \mid \omega, \sigma)$, integrating out $\omega_{1:J}$” (Section 4 paragraph 5)
>
> > ... parameterizing the variance $\sigma_j$ ...
> Typo in parametrizing. Also, should this be $\rho_j$ instead of $\sigma_j$?
>
> We keep parameterizing as we follow American English throughout the manuscript to make sure the consistency. We have corrected $\rho_j$ for $\sigma_j$. (Section 4.2 paragraph 2)
>
> > Eq. (5) and (7)
> I would suggest to maintain consistent notation of the inputs of the ELBO function across equations.
>
> The notations in Eq. (5) and Eq. (7) are indeed used to reflect different stages of the optimization process within our expectation maximization (EM) framework:
>
> * Eq. (5) presents the ELBO in its general form, encompassing all parameters: $q(w_{1:J})$, $\omega$, $\sigma$. This encapsulates the full scope of our model's parameters as they relate to the Bayesian framework.
> * In contrast, Eq. (7) is specific to the M-step of the EM algorithm, where we focus solely on optimizing $\omega$, $\sigma$ during the M-step, following the variational approximation of $q(w_{1:J})$ obtained in the E-step.
>
> These are explained in the first sentences of the respective paragraph “E-step” and “M-step”:
> “E-step. At this stage, the latent variables $\{\omega, \sigma \}$ are held fixed.”
> “M-step. After obtaining the optimized variational approximations $q(w_{1:J})$, we proceed to maximize the ELBO with respect to the latent variables $\{\omega, \sigma\}$}: …”

---

> ### Author Response · Authors · 2024-07-29
> **Response to Reviewer 2CxD - Part 6**
>
> > Questions on proposed method
> It may be useful to remark that the latent space is as big as the nodes. It is common for latent spaces to have a dimension smaller than the object of interest. Here, the latent space keeps track of the mean and variance of the nodes, so its dimension is fixed.
> Could the authors expand on whether the proposed method can be extended to latent spaces smaller than the nodes? For example, what would happen if the latent space were as large as the images being reconstructed? Would this retrieve a similar approach to INRWild?
>
> If we consider the latent space as the parameters of the MLPs used, which consist of depths of 8 with widths of either 128 or 256, the network comprises 41,473 parameters for the largest configuration (width 256). So it is indeed muchway smaller than the dimensions of the images we reconstruct, typically 512x512 or 262,144 pixels. We have some experiments (Table 6) in the Appendix comparing reconstruction performance of different MLP sizes. It shows the performance drops when decreasing the model size in exchange for computational efficiency.
>
> As for the comparison with INRWild, our method fundamentally differs. INRWild leverages a composition of static and dynamic components to handle scene variations, such as changes in lighting or object movements, within the same framework. In contrast, our Bayesian approach utilizes latent variables to capture the common pattern across multiple nodes, providing a form of KL divergence regularization that incorporates the common pattern into individual reconstruction.
>
> We have emphasized it in the abstract “we introduce a novel INR-based Bayesian framework integrating latent variables to capture the common patterns across multiple objects under joint reconstruction.The common patterns then assist in the reconstruction of each object via latent variables, thereby improving the individual reconstruction.” and Section 4.1 Paragraph 4 “The KL divergence term $D_{KL}$ serves as a regularization constraint on the network weights, pushing the posterior $q(w_j)$ to be closely aligned with the conditional prior determined by $\{\omega, \sigma\}$, which represent the collective mean and variance of all the networks in the ensemble. The KL divergence term thus serves to couple the neural representations across networks, allowing them to inform each other.”
>
> > Maybe, it could be interesting to include some example images generated from the learned prior, since the prior is an INR itself.
>
> There are many learned priors from both our method and the comparison baselines provided in the Appendix (Figures 15, 19, 22, 24).
>
> > Could the authors expand on the difference between the proposed method and the federated learning approach FedAvg? The latent mean $\omega$ can be interpreted as a meta network, so the main advantage is in considering the variance as well?
>
> FedAvg typically involves averaging the model parameters obtained from different nodes to create a shared model initialization, which is then adapted individually by each node. While this method can provide an effective starting point, its regularization effect diminishes during individual adaptations, and simple averaging may not sufficiently capture the trends present in the data.
>
> In contrast, our method treats each weight element within the network as a variable drawn from a Gaussian distribution, with both mean and variance components considered. This probabilistic approach allows our model to adapt its parameters during each learning iteration, reflecting not just the average but the underlying distribution of the training data. By maintaining a latent variance alongside the mean, our framework is equipped to capture a more comprehensive representation of the joint reconstruction nodes, effectively preserving and utilizing statistical regularities across nodes. Moreover, this regularization approach enhances robustness to noise. The advantage of our method can be seen from Table 1, Table 2, Figure 3 and Figure 6.
>
> We have also revised the Motivation part of Section 4 to better clarify the difference by including MAML and FedAvg as examples for meta-learned initialization methods “Meta-learned initialization methods, such as FedAvg andMAML, train a meta model to capture a high-level representation, which can then be promptly adapted to individual objects. Although the meta model can effectively extract statistical regularities among multiple objects, we observe that this prior information tends to be lost during the adaptation phase. This loss occurs as the adaptation relies solely on local measurements, leading to an overfitting issue, as we will discuss in Section 5.”

---

> ### Author Response · Authors · 2024-07-29
> **Response to Reviewer 2CxD - Part 7**
>
> > Similarly, all nodes are assumed to be the same size, but this assumption is not stated clearly in the text. I would suggest including this remark. Could the proposed method entail nodes of different size? What would the main hurdles to achieve that be?
>
> We acknowledge the reviewer's point about the assumption regarding node sizes, which was not initially stated clearly. To address this, we have added the sentence "Each node is assumed to be of the same size" within the manuscript to clarify this assumption. (Section 4 paragraph 4)
>
> This same size is crucial because our model uses latent variables to track the element-wise mean and variance across the network weights. If the nodes were of different sizes, each weight element might correspond to disparate aspects of the representations, making meaningful aggregation across them problematic. Adapting our method to accommodate nodes of varying sizes would require a fundamental reevaluation of how each weight element contributes to the overall representation. This exploration, while intriguing, would constitute a distinct research endeavor beyond the scope of the current study. Thus, our current framework assumes uniform node sizes to maintain coherence in the representation and computation.
>
> > Finally, could the authors expand on the choice of using the posterior expectations instead of sampling? An advantage of Bayesian methods is to include uncertainty quantification out-of-the box, and it would be interesting to compare with diffusion-based methods.
>
> We are not sure if we understand the question correctly. We have clarifiedwill clarify our method and explained why it should not be compared with diffusion-based methods as follows: Our method maximizes the marginal likelihood of measurements conditioned on latent variables. In this process, we use variational approximation to approximate the posterior distribution of network weights. Since the variational distribution is Gaussian, after training, common treatments in Bayesian frameworks involve either sampling multiple times and then averaging the results, or using the distribution mode to produce the result (i.e., maximum a posteriori). In our work, we choose the latter approach, which yields better results. We have included a subsection titled “Final reconstruction” in Appendix B4.3.
>
> Regarding comparisons with diffusion-based methods: these methods typically necessitate a prior learned from densely sampled images, which then guides the generation from sparsely sampled reconstructions. Our study focuses on techniques that do not rely on densely sampled images for training, which is why we did not include diffusion methods in our comparisons.
>
> > Missing reference for SIRT:
> Gilbert, Peter. "Iterative methods for the three-dimensional reconstruction of an object from projections." Journal of theoretical biology 36.1 (1972): 105-117.
>
> We have included the reference for SIRT.
>
> > Typo in "for the rest experiments"
>
> We have modified it to "for the rest of the experiments" (Section 5 paragraph 3)
>
> > CelebA experiments:
> It is unusual to see the CelebA dataset with a CT forward model. Have the authors considered showcasing their proposed method for a different, more common task on the CelebA dataset, such as image denoising or inpainting? This may better place the proposed method in comparison with existing denoising techniques that also do not rely on the availability of large datasets.
>
> We acknowledge the unusual application of the CelebA dataset in the context of CT reconstruction. Our initial inclusion of CelebA aimed to illustrate the potential of our method to the broader machine learning community, as TMLR is a prominent platform in this field. However, we recognize that our prior assertion of "broader application" might have overstated the method's adaptability. Consequently, we have removed the subsection discussing "broader application" to maintain the focus strictly on CT reconstruction, as indicated by the manuscript's title and content.
>
> Regarding the exploration of image denoising or inpainting, these tasks, while related, pose distinct challenges compared to CT reconstruction. We consider investigating the extension of our method to these areas as a valuable direction for future research.

---

> ### Author Response · Authors · 2024-07-29
> **Response to Reviewer 2CxD - Part 8**
>
> > Clearly state if results are on noiseless reconstruction
> CT reconstruction is presented as a noisy inverse problem. However, Table 1 shows results for noiseless measurements. This should be stated clearly before presenting results.
> Could the authors expand on their motivation for comparing on all settings for noiseless measurements and only on the inter-object setup for noisy measurements?
>
> We have now explicitly noted in the manuscript text (Section 5.1 paragraph 1), as well as in the captions of Table 1 and relevant figures (Figure 3, Figure 6), that the results pertain to noiseless CT reconstructions.
>
> Regarding the focus on noiseless measurements for most experiments, this choice was driven by empirical observations. Our baseline comparison methods demonstrated significantly poorer performance in the presence of noise, largely attributable to their propensity for overfitting, as detailed in the manuscript (Section 5 subsection overfitting). Given that this appears regardless of experiment settings and mainly related to overfitting, we only conducted inter-object experiments for noisy measurements.
>
> > ... overfitting when applied to limited data.
> Could the authors expand on this claim? Iterative reconstruction methods do not require a trained predictor, so what does "limited data" refer to here? What kind of overfitting is being referred here since measurements are noiseless at this point in the submission. In a noiseless setting, this may be an issue of stability and convergence of the optimization process rather than overfitting?
>
> We acknowledge the reviewer's concern regarding the terminology used in describing the performance deterioration of iterative methods. In CT reconstruction, traditional iterative methods like SIRT often reach a near-optimal solution relatively early during iterations, only to subsequently diverge—this phenomenon is generally recognized as a convergence issue. Conversely, within the machine learning context, especially when dealing with neural networks, overfitting refers to a model fitting too closely to the training data specifics, leading to a degradation in actual performance (in our case the actual performance is measure by the similarity to the ground truth reconstruction) despite a decrease in training loss.
>
> In our study, we examine both traditional iterative methods and learning-based INR approaches. We realize the potential for confusion due to the different implications of "convergence issues" and "overfitting" across these methods. To clarify, we have expanded our manuscript to explicitly distinguish these phenomena: "In the context of CT reconstruction, conventional iterative methods like SIRT typically approach an good approximation to the exact solution early in iterations and subsequently diverge from it, often attributed to convergence issues [Elfving et al., 2014]. In contrast, in the machine learning domain, overfitting refers to a scenario where a neural network excessively fits training data, leading to a drop in actual performance despite decreasing training loss [Srivastava et al., 2014]. Our experiments contain both non-learning-based methods (SIRT) and learning-based methods (INR). We acknowledge the subtle differences between convergence challenges and overfitting but use the term \texttt{overfitting} broadly to describe performance deterioration in both noiseless and noisy scenarios." (Footnote page 10)
>
> > It is never specified which method the curves in Figure 7 are generated with. Similarly to my confusion above, Figure 7 is used as evidence of overfitting in the noisy setting, but similar claims are also made with Figure 4 in the noiseless setting.
>
> We have mentioned now explicitly that they are from the SingleINR experiments both in text and in caption. (Figure 7)

---

> ### Author Response · Authors · 2024-07-29
> **Response to Reviewer 2CxD - Part 9**
>
> > ... we select 5 consecutive slices from new objects, choosing slices from a similar location the prior has been trained.
> It is not specified how many times this process was repeated to obtain confidence intervals. What does "similar location" mean? Is there a particular tolerance that was used to include slices from different objects?
>
> To clarify, the term "similar location" refers to the similar axial slice index, ensuring that the slices chosen for testing are anatomically comparable across different patients. Specifically, we selected 10 patients from the Decathlon Segmentation challenge dataset, each with slice numbers ranging between 220 and 240, covering similar regions of the upper body.
>
> For instance, if the prior was trained on slice number 100 across multiple patients, we selected slices 98-102 from a new patient to validate the prior-guided reconstruction. This method ensures that the slices are anatomically comparable and that the learned prior is applicable across similar anatomical regions.
>
> We have explicitly mentioned this in our manuscript: "similar location (i.e. the similar axial slice index)" to better clarify the term "similar location". (Section 5.1 subsection “Applying to unseen data using learned prior”)
>
> > Broader applicability
> I assume this paragraph is talking about the CelebA dataset, although it is never defined what "natural RGB images" are considered, beyond mentioning "faces". If this paragraph is about the CelebA experiment, I would suggest moving it before Table 1 is presented because it includes details about how those experiments are carried out.
>
> As mentioned in the previous response, we agree that the term "broader applicability" might have overstated the extension of our method to natural RGB images. To clarify, we have completely removed the subsection titled "Broader Applicability" to avoid confusion about the primary scope of our research. Instead, we have integrated the discussion about the CelebA dataset into the "Reconstruction Performance" subsection.

---

> > ### Comment · Reviewer_2CxD · 2024-07-29
> > **Thank you for your response!**
> >
> > I sincerely thank the authors for their considerations of all my questions and points of confusion, which have been addressed, and the revised version of the manuscript presents the contributions of the paper in a much clearer and relaxed way.
> >
> > I changed my original evaluation to "Claims and evidence: Yes".
> >
> > Minor typo in the footnote on Page 10, "an good approximation".
> >
> > ---
> >
> > Following up on the comment
> >
> > > Q: Finally, could the authors expand on the choice of using the posterior expectations instead of sampling? An advantage of Bayesian methods is to include uncertainty quantification out-of-the box, and it would be interesting to compare with diffusion-based methods.
> >
> > > A: We are not sure if we understand the question correctly. We have clarifiedwill clarify our method and explained why it should not be compared with diffusion-based methods as follows: Our method maximizes the marginal likelihood of measurements conditioned on latent variables. In this process, we use variational approximation to approximate the posterior distribution of network weights. Since the variational distribution is Gaussian, after training, common treatments in Bayesian frameworks involve either sampling multiple times and then averaging the results, or using the distribution mode to produce the result (i.e., maximum a posteriori). In our work, we choose the latter approach, which yields better results. We have included a subsection titled “Final reconstruction” in Appendix B4.3. Regarding comparisons with diffusion-based methods: these methods typically necessitate a prior learned from densely sampled images, which then guides the generation from sparsely sampled reconstructions. Our study focuses on techniques that do not rely on densely sampled images for training, which is why we did not include diffusion methods in our comparisons.
> >
> > I agree with the authors regarding the comparison with diffusion-based models, and I thank them for clarifying how the final reconstruction is obtained in Appendix B.4.
> >
> > The point I was trying to make was that a different way to obtain reconstructions could be to sample the network parameters, reconstruct, sample again, then reconstruct, and so on and so forth several times. Then, one could report both the average and variance of the network outputs in order to obtain a surrogate notion of uncertainty in the learned nodes.
> >
> > This could be an advantage of a Bayesian INR framework---which provides a distribution over parameters compared to a point-predictor---as uncertainty quantification is important for high-stakes scenarios such as CT reconstruction, and it has been gaining attention in the broader machine learning and medical imaging community.
> >
> > For example:
> >
> > [1] Gal, Yarin, and Zoubin Ghahramani. "Dropout as a bayesian approximation: Representing model uncertainty in deep learning." (2016)
> >
> > [2] Angelopoulos, Anastasios N., et al. "Image-to-image regression with distribution-free uncertainty quantification and applications in imaging." (2022)

---

> > > ### Author Response · Authors · 2024-07-31
> > > **Response to Reviewer 2CxD**
> > >
> > > Thank you for your clarification and continued engagement with our manuscript! We have corrected the typo from "an good approximation" to "a good approximation". Regarding uncertainty quantification, we agree that it is a very relevant and interesting topic, particularly in the context of medical imaging where Bayesian methods can provide meaningful insights into the reliability of reconstructions.
> > >
> > > To address this, we have included a new subsection in Appendix D.10 of our manuscript. This section demonstrates how our proposed method, INR-Bayes, can potentially be extended to include uncertainty quantification:
> > > “We employ a Bayesian framework to identify common patterns among jointly reconstructed objects. While the primary focus of this study is not on uncertainty quantification [Gal & Ghahramani, 2016; Angelopoulos et al.,2022], we demonstrate how \verb|INR-Bayes| can facilitate this aspect within CT reconstructions. During training, each node approximates the posterior distribution of its weights as a Gaussian distribution. This approach allows us to sample the network parameters multiple times after training to quantify uncertainty. To illustrate this, we sample the network parameters ten times to generate ten distinct reconstructions in an intra-object setup using the LungCT dataset. We then compute the mean and variance of these reconstructions to show the expected reconstruction and characterize the associated uncertainty in Figure 26.”

---

### Review · Reviewer_k9NV · 2024-07-18

**Summary Of Contributions:**

In this submission, the authors proposed INR-Bayes, a Bayesian-based joint reconstruction method for CT imaging. They proposed to learn neural representation of multiple object jointly, and the neural network weights are sampled from a multivariate Gaussian. To optimize the likelihood, they maximize the evidence lower bound using the EM algorithm. The authors demonstrated their method’s effectiveness both qualitatively and quantitatively.

**Audience:**

Yes

**Broader Impact Concerns:**

No ethical concerns.

**Claims And Evidence:**

Yes

**Requested Changes:**

* SingleINR and MAML are strong baselines. It would be great if we have more comparisons to them and can clearly demonstrate INR-Bayes's advantages over them.
* More comparisons between inter and intra-object joint reconstructions. Is it better to optimize on different views of a single object, or different views of multiple objects? Note that we can have more slices when optimizing on multiple objects. What would be the difference of the learned priors?

**Strengths And Weaknesses:**

+ The paper is clearly written. Necessary and detailed background information is provided in the paper, so that the reader can understand the idea easily.
+ The paper contains detailed comparisons between INR-Bayes and baseline algorithms.
- Checking Table 1 and Figure 3 we can see SingleINR and MAML are the strongest baselines. More comparisons can be done to prove INR-Bayes is better than these two.

---

> ### Author Response · Authors · 2024-07-29
> **Response to Reviewer k9NV**
>
> Thank you for your detailed review and positive feedback on the clarity and comprehensiveness of our manuscript. We are encouraged by your recognition of the detailed comparisons provided and the background information that aids in understanding our method. Below, we respond to your specific feedback:
>
> > SingleINR and MAML are strong baselines. It would be great if we have more comparisons to them and can clearly demonstrate INR-Bayes's advantages over them.
>
> We appreciate the suggestion to enhance comparisons with strong baselines like SingleINR and MAML. In response, we have expanded our experimental section to include additional comparisons across various setups, including the inclusion of established regularization-based methods aligning with the request from reviewer bPhg. (Table 1, Table 2, Figure 3, Figure 6, Section 5.1 subsection “Reconstruction Performance”)
>
> Our method consistently demonstrates superior performance in terms of reconstruction quality, outperforming comparison methods in nearly all tested scenarios, except for a case on the CelebA dataset where a regularization-based method achieved the highest SSIM. Notably, under noisy conditions, our INR-Bayes method maintains robust performance, whereas the baselines exhibit significant degradation. Additionally, in scenarios involving adaptation to new objects and varying scanning angles, INR-Bayes continues to excel compared to these baselines.
>
> We believe these comprehensive evaluations adequately demonstrate the advantages of INR-Bayes over the established baselines. However, we are open to specific suggestions regarding further experiments that might enhance this comparison.
>
> > More comparisons between inter and intra-object joint reconstructions. Is it better to optimize on different views of a single object, or different views of multiple objects? Note that we can have more slices when optimizing on multiple objects. What would be the difference of the learned priors?
>
>
> We evaluated the reconstruction of 10 target slices across four different configurations: intra-object (1x10), inter-object (10x1), and two intermediate settings—2x5 and 5x2. Our results indicate that the intermediate configurations typically yield slightly better PSNR and SSIM scores compared to purely inter- or intra-object configurations. This suggests that for optimal reconstruction quality of specific objects, a balanced blend of similarity and diversity among the joint reconstruction objects can be crucial. Excessive similarity (as in intra-object settings) or diversity (as in inter-object settings) tends to diminish performance. Regarding the learned priors, the intra-object configuration most closely approximates the target object, whereas the inter-object configuration shows the greatest deviation. These findings are detailed in Section D.6 of the appendix.

---

### Decision · Action_Editor_CH9r · 2024-09-02

**Recommendation:** Accept as is

**Comment:**

The consensus is clearly that the paper should be accepted.
The only outstanding comment is  the narrow scope of the paper. The reviewers comments  do not imply that certifications should be awarded.

**Audience:**

Some of the TMLR audience will be interested in the paper, but the scope of the  paper is actually pretty narrow within the TMLR community.

**Claims And Evidence:**

The submission introduces a novel Bayesian approach for the joint reconstruction of several objects from sparse-view CT measurements. The technical novelty consists of training object-specific implicit neural representations (INRs) concurrently by coupling them with a latent prior that should encode common information across objects.
Experiments and ablation tests show that the proposed method outperforms alternatives on 4 different datasets coming from different domains.
After the reviews, the authors have added more ablation studies and comparisons, providing strong evidence in favor of their claims.